# UNIFEWS: You Need Fewer Operations for Efficient Graph Neural Networks

**Ningyi Liao** [1]  **Zihao Yu** [1]  **Ruixiao Zeng** [1]  **Siqiang Luo** [1]

## Abstract

Graph Neural Networks (GNNs) have shown promising performance, but at the cost of resource-intensive operations on graph-scale matrices. To reduce computational overhead, previous studies attempt to sparsify the graph or network parameters, but with limited flexibility and precision boundaries. In this work, we propose UNIFEWS, a joint sparsification technique to unify graph and weight matrix operations and enhance GNN learning efficiency. The UNIFEWS design enables adaptive compression across GNN layers with progressively increased sparsity, and is applicable to a variety of architectures with on-the-fly simplification. Theoretically, we establish a novel framework to characterize sparsified GNN learning in view of the graph optimization process, showing that UNIFEWS effectively approximates the learning objective with bounded error and reduced computational overhead. Extensive experiments demonstrate that UNIFEWS achieves efficiency improvements with comparable or better accuracy, including $10-20\times$ matrix operation reduction and up to $100\times$ acceleration for graphs up to billion-edge scale. Our code is available at: https://github.com/gdmnl/Unifews.

## 1. Introduction

Graph Neural Networks (GNNs) have undergone extensive development for learning from graph-structure data and have achieved remarkable performance (Kipf & Welling, 2017; Hamilton et al., 2017; Veličković et al., 2018). The power of GNNs is mainly attributed to the message-passing scheme containing two alternative operations, namely *graph propagation* and *feature transformation*, which aggregates neighborhood information in the graph and updates representation by trainable weights.

Despite their success, canonical GNNs are recognized for their high resource consumption, especially when scaling up to large graphs (Wu et al., 2021). The performance bottleneck arises from the connection with graph size, as both graph propagation and feature transformation can be viewed as multiplications on graph-scale matrices, and lead to computational complexities proportional to the numbers of edges and nodes in the graph, respectively (Chen et al., 2020). Hence, the essence of improving GNN learning efficiency lies in reducing the number of operations associated with graph diffusion and model weights (Liu et al., 2022; Zhang et al., 2023a).

In efforts to reduce the *graph* part of computation, prior GNN graph sparsification techniques (Liu et al., 2019) typically remove graph components based on either predetermined (Zheng et al., 2020; Zeng et al., 2021; Liu et al., 2023b) or learned (Li et al., 2021; 2022; Jin et al., 2022) criteria. Another direction is to exploit compression on the *model* architecture by integrating network pruning approaches (Deng et al., 2020a), which gradually sparsifies weight elements during GNN training (Zhou et al., 2021; Chen et al., 2021; You et al., 2022; Wang et al., 2023b). Both approaches, however, suffer from limited flexibility due to the static graph structure throughout all propagation iterations: the topology may become overly coarse, thereby omitting crucial information for GNN feature extraction, or may be under-sparsified and model efficiency is scarcely improved. Moreover, there is a noticeable gap in the *theoretical* interpretation of GNN sparsification. Existing works are either constrained to the layer-agnostic graph approximation (Chen et al., 2018; Zou et al., 2019), or only provide straightforward representation error analysis (Srinivasa et al., 2020; Liu et al., 2023b). The impact of simplified graphs on the GNN learning process, particularly through multiple layers, remains inadequately addressed.

To mitigate the drawbacks of current GNN compression, we propose **UNIFEWS**, a UNIFied Entry-Wise Sparsi-fication framework for GNN graph and weights. As illustrated in Figure 1, by considering the two stages in GNN as matrix operations, graph and model sparsification can be unified in an entry-wise manner, i.e., manipulating individual elements during matrix multiplication. We then establish a theoretical framework to quantify the impact of UNIFEWS on precision and efficiency. By bridging the graph learning process in

---

[1]College of Computing and Data Science, Nanyang Technological University, Singapore. Correspondence to: Siqiang Luo <siqiang.luo@ntu.edu.sg>.

*Proceedings of the $42^{nd}$ International Conference on Machine Learning*, Vancouver, Canada. PMLR 267, 2025. Copyright 2025 by the author(s).

the spectral domain, our analysis elucidates the effect of joint sparsification across iterative GNN layers in a holistic manner, which differs from previous approaches focusing on a particularized metric for approximation.

In practice, UNIFEWS is capable of reducing operations in the two-stage GNN message-passing and improving efficiency. The simple but effective strategy is powerful in conducting flexible computation for each node and each GNN layer, effectively balancing the operational reduction while retaining precision. The theoretical and implementation framework of UNIFEWS is applicable to a broad family of GNNs, covering the representative models of both iterative and decoupled architectures.

In summary, (1) we propose UNIFEWS for unifying GNN graph and model sparsification from an entry-wise perspective to reduce graph-scale operations and alleviate computational cost; (2) we develop novel theoretical framework bridging graph spectral filtering and sparsification as approximation; (3) we experimentally demonstrate that UNIFEWS outperforms existing GNN compressions by reaching over 90% of joint sparsity without compromising accuracy. On the billion-scale graph papers100M, UNIFEWS is able to boost the computation time by $100\times$.

## 2. Preliminaries and Related Works

### 2.1. GNN Architecture

**Graph Notation.** Consider a graph $\mathcal{G} = \langle \mathcal{V}, \mathcal{E} \rangle$ with $n = |\mathcal{V}|$ nodes and $m = |\mathcal{E}|$ edges. The neighborhood set of a node $u \in \mathcal{V}$ is $\mathcal{N}(u) = \{v \mid (u,v) \in \mathcal{E}\}$, and its degree $d(u) = |\mathcal{N}(u)|$. The self-looped adjacency matrix is $\bar{A} \in \mathbb{R}^{n \times n}$ and the diagonal degree matrix is $D = \text{diag}(d(u_1), d(u_2), \cdots, d(u_n))$. The normalized adjacency and Laplacian matrices are $\tilde{A} = D^{-1/2} \bar{A} D^{-1/2}$ and $\tilde{L} = I - \tilde{A}$, respectively.

**Iterative GNN.** Iterative models such as GCN (Kipf & Welling, 2017) take the node attribute $X \in \mathbb{R}^{n \times f}$ as input, and recurrently update by applying the graph diffusion matrix $T$. The representation matrix $H_{(l+1)}$ of the $(l+1)$-th layer is updated as $H_{(l+1)} = \sigma\left(T H_{(l)} W_{(l)}\right)$, where $l \in \{0, \cdots, L-1\}$, and $\sigma(\cdot)$ denotes the activation function. It implies two consecutive steps, that *graph propagation* computes the embedding $P_{(l)} = T H_{(l)}$, and *feature transformation* multiplies the learnable weight $W_{(l)}$. For example, the diffusion matrix of GCN is $T = \tilde{A}$.

**Decoupled GNN.** This variant simplifies GNN architecture by separating the propagation from iterative updates (Wu et al., 2019; Gasteiger et al., 2019; Wang et al., 2021a). The graph-related propagation is computed in advance as the embedding matrix $P_{(l+1)} = T_{(l)} \cdot P_{(l)}$ for $l \in \{0, \cdots, L-1\}$, and the transformation is as simple as an $L$-layer MLP that $H_{(l+1)} = \sigma\left(H_{(l)} W_{(l)}\right)$, where the boundary conditions are $P_{(0)} = X$ and $H_{(0)} = P_{(L)}$.

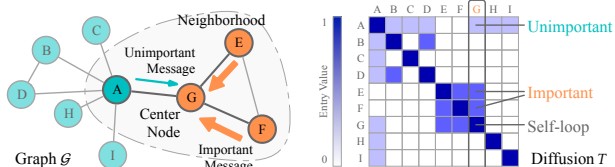

(a) An example graph and the corresponding diffusion matrix.

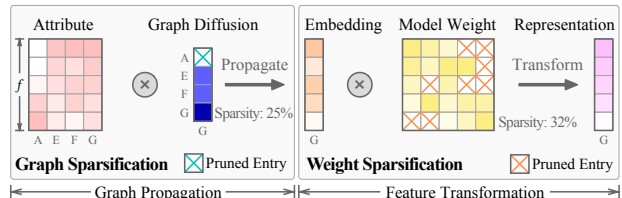

(b) UNIFEWS sparsification on graph and model entries.

Figure 1: **(a)** Graph propagation messages from neighbors with larger diffusion values are associated with greater importance. **(b)** UNIFEWS jointly applies sparsification to both GNN stages. Unimportant entries of the diffusion and weight matrices are pruned to reduce operations.

As an exemplar, the diffusion matrices for SGC (Wu et al., 2019) and APPNP (Klicpera et al., 2019) are $T = \tilde{A}$ and $T = \alpha(1-\alpha)\tilde{A}$ according to their respective designs.

### 2.2. Taxonomy of GNN Sparsification

**Iterative Graph Sparsification.** Graph sparsification methods aim to augment the iterative message-passing and improve accuracy (Liu et al., 2022). NeuralSparse (Zheng et al., 2020) and LSP (Kosman et al., 2022) drop edges based on topological properties, while AdaptiveGCN (Li et al., 2021) and SGCN (Li et al., 2022) implement learnable graph connections. FastGAT (Srinivasa et al., 2020) and DSpar (Liu et al., 2023b) specifically examine spectral patterns. However, these techniques modify the graph in a one-time and layer-agnostic manner, resulting in limited flexibility for balancing efficiency and efficacy.

**Decoupled Propagation Personalization.** A particularized branch emerges for simplifying decoupled GNNs (Zhang et al., 2021; Huang et al., 2023; Liao et al., 2023b; Zeng et al., 2021). Since the graph operation is isolated, it is more flexible for fine-grained modification, which replaces the graph-level message-passing with separate maneuver on the edge connection for each propagation hop as in Figure 2(a). Our UNIFEWS is akin to this idea, but with a better suitability for both iterative and decoupled propagations.

**Architectural and Joint Compression.** A series of research take the network architecture into account. Their general pipeline, as outlined in Figure 2(b), involves a model compressor and a static graph sparsifier. Some works treat the two stages independently (Zhou et al., 2021; Liu et al., 2023a), while others exploit joint updates (You et al., 2022; Sui et al., 2024; Chen et al., 2021; Wang et al., 2023b;a). One critical drawback of these methods lies in the additional

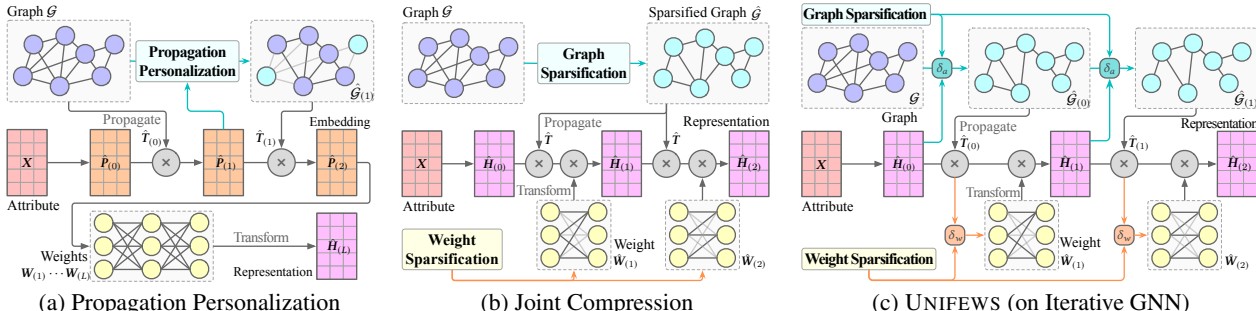

(a) Propagation Personalization      (b) Joint Compression      (c) UNIFEWS (on Iterative GNN)

Figure 2: Comparison of GNN pipelines between conventional simplification techniques and our UNIFEWS framework. **(a)** Personalized propagation iteratively simplifies the node-dependent graph diffusion but is only applicable to decoupled GNNs. **(b)** Joint model compression can sparsify both graph and weight, whereas the same diffusion is utilized across all layers. **(c)** UNIFEWS implements entry-wise sparsification of both graph and weights for each layer with increased sparsity.

bottleneck for learning the compression strategy, which still incurs full-scale computation and are less efficient. A thorough discussion regarding other efficient GNNs such as graph sampling can be found in Appendix A.

## 3. Graph Smoothing under Sparsification

In this section, we first relate GNN learning with the graph smoothing process, which enables us to characterize the process by our novel approximation bound. Generally, a broad scope of both iterative and decoupled GNNs can be interpreted by a spectral graph smoothing process (Zhu et al., 2021). We adopt the optimization framework as:

**Definition 3.1** (**Graph Laplacian Smoothing** (Ma et al., 2021)). Given a weighted graph $\mathcal{G} = \langle \mathcal{V}, \mathcal{E} \rangle$ with Laplacian matrix $\boldsymbol{L}$. Based on an input signal vector $\boldsymbol{x} \in \mathbb{R}^n$, the Graph Laplacian Smoothing problem aims to optimize vector $\boldsymbol{p} \in \mathbb{R}^n$ with the goal:

$$\boldsymbol{p}^* = \arg\min_{\boldsymbol{p}} \mathcal{L}, \quad \mathcal{L} = \|\boldsymbol{p} - \boldsymbol{x}\|^2 + c \cdot \boldsymbol{p}^\top \boldsymbol{L} \boldsymbol{p}, \quad (1)$$

where $\| \cdot \|$ is the vector $L_2$ norm and the regularization coefficient $c$ is chosen from $[0, 1]$.

In Equation (1), $\boldsymbol{L}$ is the general Laplacian matrix, as normalization only causes a difference in coefficient. The first term of $\mathcal{L}$ reflects the closeness to the *input signal*, i.e., node attributes representing their identity. The second term is associated with the *graph structure*, acting as a regularization that constrains the representation values of connected node pairs to be similar. The closed-form solution $\boldsymbol{p}^* = (\boldsymbol{I} + c\boldsymbol{L})^{-1}\boldsymbol{x}$ can be inferred when the derivative $\partial \mathcal{L}/\partial \boldsymbol{p} = 0$. However, note that it is prohibitive to directly acquire the converged solution due to the inverse calculation on large graph-scale matrix. Hence, GNN models employ an iterative approach to learn the representation under this framework with varying architectural designs. For example, GCN convolution corresponds to $c = 1$.

Next, we introduce graph sparsifiers as an approximation process of graph smoothing. In order to measure the extend

of approximation and its impact on learning outcomes, we consider the spectral similarity:

**Definition 3.2** ($\epsilon$-**Spectral Similarity**). The approximate Laplacian matrix $\hat{\boldsymbol{L}}$ is said to be $\epsilon$-spectrally similar to the raw Laplacian matrix $\boldsymbol{L}$ if:

$$\boldsymbol{x}^\top(\hat{\boldsymbol{L}} - \epsilon\boldsymbol{I})\boldsymbol{x} \le \boldsymbol{x}^\top \boldsymbol{L}\boldsymbol{x} \le \boldsymbol{x}^\top(\hat{\boldsymbol{L}} + \epsilon\boldsymbol{I})\boldsymbol{x}, \ \forall \boldsymbol{x} \in \mathbb{R}^n. \ (2)$$

Graph approximation satisfying Definition 3.2 can be regarded as spectral sparsification (Spielman & Srivastava, 2011; Batson et al., 2013), which identifies operations that maintains certain spectral properties such as eigenvalues during modification. Compared to the common *multiplicative* spectral similarity (Sadhanala et al., 2016; Calandriello et al., 2018; Charalambides & Hero, 2023), Definition 3.2 possesses an *additive* tolerance, which allows for manipulating specific entries of $\boldsymbol{L}$ and suits our scenario.

Then, we are able to characterize the graph smoothing problem for sparsified graphs. Under edge modifications, we intend to bound the optimization goal under approximation:

**Lemma 3.3** (**Approximate Graph Laplacian Smoothing**). *Given two graphs $\mathcal{G} = \langle \mathcal{V}, \mathcal{E} \rangle$ and $\hat{\mathcal{G}} = \langle \mathcal{V}, \hat{\mathcal{E}} \rangle$, where $\hat{\mathcal{E}}$ is the sparsified edge set. When the Laplacian $\hat{\boldsymbol{L}}$ of $\hat{\mathcal{G}}$ is $\epsilon$-similar to $\boldsymbol{L}$ of $\mathcal{G}$, the solution $\hat{\boldsymbol{p}}^*$ to the problem Equation (1) w.r.t $\hat{\boldsymbol{L}}$ is called an $\epsilon$-approximation of the solution $\boldsymbol{p}^*$ w.r.t. $\boldsymbol{L}$, and:*

$$\|\hat{\boldsymbol{p}}^* - \boldsymbol{p}^*\| \le c\epsilon\|\boldsymbol{p}^*\|. \quad (3)$$

The proof of Lemma 3.3 can be found in Appendix B.1. It establishes a novel interpretation for characterizing the smoothing procedure under a sparsified graph, that if a sparsifier complies with the spectral similarity Definition 3.2, it is capable of effectively approximating the iterative graph optimization and achieving a close output with bounded error. Compared to approximation bounds specifying feature values in previous GNN sparsification theories (Srinivasa et al., 2020; Zhang et al., 2023b), our analysis highlights the impact of graph sparsifier throughout the holistic learning process, which enjoys better expressiveness and suitability.

# 4. UNIFEWS Algorithm and Theory

This section presents our UNIFEWS framework by respectively developing its application to decoupled and iterative GNNs, where graph and weight sparsification are separately performed in the former architecture, and are combined in the latter. Our key theoretical insights are summarized as:

• **Bridging sparsification and smoothing:** UNIFEWS can be described as an spectral sparsifier. Its sparsification strength is determined by the threshold of entry removal.

• **Multi-layer bounds:** UNIFEWS provides an approximation to the graph smoothing optimization. Its representation error to the original objective is effectively bounded.

• **Improvements on efficiency and efficacy:** UNIFEWS is advantageous in enhancing efficiency, mitigating the oversmoothing issue, and facilitating enhanced joint sparsity.

## 4.1. UNIFEWS as Spectral Sparsification

**Intuition: Entry Values Denote Importance in Graph Computation.** As depicted in Figure 2(b), conventional graph sparsification methods for GNNs only provide fixed and uniform *graph-level* adjustments through the entire learning process. This lack of flexibility hinders their performance when employing to the recurrent update design with multiple GNN layers. Contrarily, *node-wise* models in Figure 2(a) aim to personalize the graph-based propagation, but come with additional explicit calculations and impaired runtime efficiency. We are hence motivated to design a sparsification approach that (1) enables modifications in a *fine-grained* manner to further simplify the computation; and (2) can be seamlessly integrated into the matrix operation with *negligible overhead*.

To this end, we first focus on the isolated graph propagation stage and express the propagation of decoupled models in an *entry-wise* manner on node $u$ as:

$$\boldsymbol{p}_{(l+1)}[u] = \sum_{v \in \mathcal{N}(u)} \tau_{(l)}[u,v] = \sum_{v \in \mathcal{N}(u)} \boldsymbol{T}_{(l)}[u,v] \cdot \boldsymbol{p}_{(l)}[v], \quad (4)$$

where the propagation message $\tau_{(l)}[u,v]$ from node $v$ to $u$ is regarded as an entry. As illustrated in Figure 1(a), in a single hop of GNN diffusion, edges carrying propagation messages exert varying impacts on neighboring nodes, whose importance is dependent on the graph topology. Messages with minor significance are usually considered redundant and can be eliminated to facilitate sparsity. From the perspective of matrix computation, this is achieved by omitting the particular message $\tau[u,v]$ in current aggregation based on an importance indicator such as its magnitude.

**Edge Pruning for Single Layer.** In order to perform pruning on entry $\tau[u,v]$, the diffusion matrix $\boldsymbol{T}$ can be utilized to record the pruning information, which is exactly the concept of graph sparsification. Given a universal magnitude threshold $\delta_a$, zeroing out entries $|\tau[u,v]| < \delta_a$ is equivalent

to sparsifying $\boldsymbol{T}$ with a node-wise threshold $\delta_a'$ as:

$$\hat{\boldsymbol{T}}[u,v] = \mathrm{thr}_{\delta_a'}\big(\boldsymbol{T}[u,v]\big) \cdot \boldsymbol{T}[u,v], \quad \delta_a' = \delta_a/|\boldsymbol{p}[v]|, \quad (5)$$

where the pruning function with an arbitrary threshold $\delta$ is $\mathrm{thr}_\delta(x) = 1$ if $|x| > \delta$, and $\mathrm{thr}_\delta(x) = 0$ otherwise. Sparsification by Equation (5) has two concurrent effects: for graph computation, messages with small magnitudes are prevented from propagating to neighbors and composing the output representation; for graph topology, corresponding edges are removed from the diffusion process.

Next, we show that edge pruning in Equation (5) for one layer can be considered as a spectral sparsification following Definition 3.2, which enables us to derive its approximation bound. Practically, the target sparsity $\eta_a \in [0,1]$ is usually determined by the realistic application. Let $\hat{\mathcal{E}}$ be the edge set achieved by UNIFEWS sparsification and the corresponding Laplacian is $\hat{\boldsymbol{L}}$, there is $\eta_a = 1 - |\hat{\mathcal{E}}|/m$. We first derive the following lemma as a general guarantee associating sparsification and spectral similarity:

**Lemma 4.1.** *Given graph $\mathcal{G}$ and embedding $\boldsymbol{p}$, let $\hat{\mathcal{E}}$ be the edge set achieved by graph sparsification with threshold $\delta_a$. The sparsified Laplacian matrix $\hat{\boldsymbol{L}}$ is $\epsilon$-similar to $\boldsymbol{L}$ when $q_a \delta_a \leq \epsilon \|\boldsymbol{p}\|$, where $q_a$ is the number of edges removed.*

The proof of Lemma 4.1 is detailed in Appendix B.2. To derive more specific bounds of $q_a$, we need to examine the actual distribution of entry values. Regarding the assumption on edge distribution, we particularly investigate the scale-free graph, which is common in realistic large-scale graphs (Clauset et al., 2009). For input attribute values, we consider the Gaussian distribution, which is also commonly used for depicting the feature distribution of neural networks as an extension of the central limit theorem (Deshpande et al., 2018; Bojchevski & Günnemann, 2018). With these two assumptions, the following theorem relates the threshold and approximation bound with sparsification:

**Theorem 4.2 (Bound for UNIFEWS).** *Given a graph $\mathcal{G}$, embedding $\boldsymbol{p}$, and required edge sparsity $\eta_a$, the threshold $\delta_a$ can be decided by $\delta_a = C(1 - \eta_a)^{-t}$, and the sparsified Laplacian $\hat{\boldsymbol{L}}$ is $\epsilon$-similar to $\boldsymbol{L}$ with the approximation bound:*

$$\epsilon = O\left(\eta_a (1 - \eta_a)^{-t}\right), \quad (6)$$

*where $C$ and $t$ are positive constants. $C > 0$ is determined by the embedding values $\|\boldsymbol{p}\|$, and $0 < t < 1$ depicts the degree distribution.*

The proof of Theorem 4.2 and specific derivations of $C$ and $t$ can be found in Appendix B.3. The significance of Theorem 4.2 lies in bridging the sparsification technique and Laplacian smoothing with a specific precision bound for the first time to the best of our knowledge. Its implication is that, the adaptive sparsification by UNIFEWS for a single layer qualifies as a graph spectral sparsifier bounded by $\epsilon$. Its pivotal parameter, i.e., the sparsification threshold,

and its approximation bound can be determined once given the sparsity $\eta_a$. A sparsity closer to $1$ results in a larger threshold value as well as a loose error guarantee.

## 4.2. UNIFEWS for Graph Propagation

Thanks to the adaptive property of the entry-wise pruning scheme, we are able to further perform fine-grained and gradual sparsification across propagation layers. Intuitively, a message deemed of minor significance in the current propagation is unlikely to propagate further and influence more distant nodes in subsequent layers. UNIFEWS hence offers progressively increasing sparsity throughout GNN layers.

As depicted in Figure 2(c), UNIFEWS iteratively applies sparsification to each layer, and the pruned diffusion matrix $\hat{T}_{(l)}$ is inherited to the next layer for further edge removal. Denote the edge set corresponding to $\hat{T}_{(l)}$ as $\hat{\mathcal{E}}_{(l)}$, there is $\hat{\mathcal{E}}_{(l)} \subseteq \hat{\mathcal{E}}_{(l-1)} \subseteq \cdots \subseteq \hat{\mathcal{E}}_{(0)} \subseteq \mathcal{E}$, which indicates a diminishing number of operations for deeper layers. Consequently, Equation (4) is modified as the following to represent entry-level diffusion on the sparsified graph:

$$\hat{p}_{(l+1)}[u] = \sum_{v \in \mathcal{N}(u)} \hat{T}_{(l)}[u,v] \cdot p_{(l)}[v] = \sum_{v \in \hat{\mathcal{N}}_{(l)}(u)} \tau_{(l)}[u,v], \quad (7)$$

where the neighborhood composed by remaining connections is $\hat{\mathcal{N}}_{(l)}(u) = \{v \mid (u,v) \in \hat{\mathcal{E}}_{(l)}\}$.

We illustrate the application of UNIFEWS on decoupled graph propagation in Algorithm 1 as highlighted components. Note that the nested loops starting from Line 5 are identical to the canonical graph propagation conducted by sparse-dense matrix multiplication in common GNNs. Compared to the normal propagation Line 11, it refrains the value update and prunes the corresponding edge for insignificant entries in Line 9. Hence, UNIFEWS can be implemented into the GNN computation to enhance efficiency without additional matrix-wise overhead. For multi-dimensional feature matrix, the pruning function Equation (5) is performed by replacing $|p[v]|$ with the specific norm $\|P[v]\|$ across feature dimensions. Note that we adopt skip connections to each layer in the algorithm by initializing $P_{(l+1)}$ explicitly with the value in the previous hop, in order to preserve at least one meaningful value for each node embedding in case all connected edges are pruned.

**Approximation Bound.** From the theoretical perspective, the error introduced by multi-hop propagation is more complicated than the single-layer case in Theorem 4.2, as it is composed of both the diffusion and embedding values in approximation. By exploiting the GNN graph smoothing process in Definition 3.1, we show that:

$$\|\hat{p}_{(l+1)} - p_{(l+1)}\| \leq \|\hat{p}_{(l)} - p_{(l)}\| + O(\epsilon) \cdot \|\hat{T} - T\|_2 \|p_{(l)}\|.$$

Therefore, as long as the sparsification satisfies Theorem 4.2 for each hop, the whole process of pruning and accumulating the sparsified graph across multiple hops can be regarded as the approximate smoothing in Lemma 3.3, as stated in the following proposition and detailed in Appendix B.4:

**Proposition 4.3** (**Progressive UNIFEWS for graph propagation**). *For an $L$-hop graph propagation under Algorithm 1, if the final sparsifier satisfies $\epsilon$-similarity, then the overall process is an $\epsilon$-approximation to the original graph smoothing problem.*

Moreover, the feature transformation stage in decoupled GNNs is feasible to the weight pruning described in the following section, which is similar to classical irregular pruning with magnitude-based thresholds for MLP networks (Han et al., 2015; Srinivas & Babu, 2015).

---

**Algorithm 1:** UNIFEWS on Decoupled Propagation

**Input:** Graph $\mathcal{G} = \langle \mathcal{V}, \mathcal{E} \rangle$, diffusion $T_{(l)}$, attribute $X$, propagation hop $L$, graph sparsification threshold $\delta_a$
**Output:** Approximate embedding $\hat{P}_{(L)}$

1  $\hat{P}_{(0)} \leftarrow X$, $\hat{\mathcal{E}}_{(-1)} \leftarrow \mathcal{E}$
2  **for** $l = 0$ to $L - 1$ **do**
3   $\hat{P}_{(l+1)} \leftarrow \hat{P}_{(l)}$
4   $\hat{\mathcal{E}}_{(l)} \leftarrow \hat{\mathcal{E}}_{(l-1)}$, $\hat{T}_{(l)} \leftarrow T_{(l)}$
5   **for all** $u \in \mathcal{V}$ **do** ▷ *[matrix op]*
6    $\hat{\mathcal{N}}_{(l)}(u) \leftarrow \{v \mid (u,v) \in \hat{\mathcal{E}}_{(l)}, v \neq u\}$
7    **for all** $v \in \hat{\mathcal{N}}_{(l)}(u)$ **do**
8     **if** $|\hat{T}_{(l)}[u,v]| < \delta_a / \|\hat{P}_{(l)}[v]\|$ **then**
9      $\hat{\mathcal{E}}_{(l)} \leftarrow \hat{\mathcal{E}}_{(l)} \setminus \{(u,v)\}$, $\hat{T}_{(l)}[u,v] \leftarrow 0$
10    **else**
11     $\hat{P}_{(l+1)}[u] \leftarrow \hat{P}_{(l+1)}[v] + \hat{T}_{(l)}[u,v] \cdot \hat{P}_{(l)}[v]$
12 **return** $\hat{P}_{(L)}$

---

**Algorithm 2:** UNIFEWS on Iterative GNN

**Input:** Graph $\mathcal{G} = \langle \mathcal{V}, \mathcal{E} \rangle$, diffusion $T_{(l)}$, attribute $X$, network layer $L$, graph sparsification threshold $\delta_a$, weight sparsification threshold $\delta_w$
**Output:** Approximate representation $\hat{H}_{(L)}$

1  $\hat{H}_{(0)} \leftarrow X$, $\hat{\mathcal{E}}_{(-1)} \leftarrow \mathcal{E}$
2  **for** $l = 0$ to $L - 1$ **do**
3   $\hat{H}_{(l+1)} \leftarrow 0$, $\hat{P}_{(l)} \leftarrow \hat{H}_{(l)}$
4   Acquire sparsified $\hat{P}_{(l)}, \hat{\mathcal{E}}_{(l)}, \hat{T}_{(l)}$ as in Algorithm 1
5   **for** $i \leftarrow 1$ to $f$ **do** ▷ *[matrix op]*
6    **for** $j \leftarrow 1$ to $f$ and $\hat{W}_{(l)}[j,i] \neq 0$ **do**
7     **if** $|\hat{W}_{(l)}[j,i]| < \delta_w / \|\hat{P}_{(l)}[:,j]\|$ **then**
8      $\hat{W}_{(l)}[j,i] \leftarrow 0$
9     **else**
10     $\hat{H}_{(l+1)}[:,i] \leftarrow \hat{H}_{(l+1)}[:,i] + \hat{W}_{(l)}[j,i] \cdot \hat{P}_{(l)}[j]$
11  $\hat{H}_{(l+1)} \leftarrow \sigma(\hat{H}_{(l+1)})$
12 **return** $\hat{H}_{(L)}$

## 4.3. UNIFEWS for Iterative Update

Compared to decoupled designs, graph propagation and feature transformation in iterative GNN models are tightly integrated in each layer. Traditionally, this poses a challenge to GNN sparsification methods depicted in Figure 2(b), as specialized schemes are necessary for simultaneously modifying the two components without impairing the performance. On the contrary, our UNIFEWS approach takes advantage of this architecture for jointly sparsifying the model towards a win-win situation in graph learning.

The sparsification of iterative UNIFEWS for the entire message-passing process is presented in Algorithm 2, where the difference from the common scheme is also highlighted . Similarly, although presented as nested loops in the algorithm, the multiplication and sparsification are implemented as matrix operations. Its graph propagation stage is identical to Algorithm 1, except that the embedding $\hat{P}_{(l)}$ is initialized by the previous representation $\hat{H}_{(l)}$. Meanwhile, sparsification for weights is relatively straightforward compared to graph edges, since weight matrices are structured and their approximation is well-studied as network pruning (Han et al., 2015; Srinivas & Babu, 2015; Deng et al., 2020a). Given the embedding of the current layer $\hat{P}_{(l)}$, we similarly rewrite the iterative GNN update as:

$$\boldsymbol{H}_{(l+1)}[:,i] = \sigma\Big(\sum_{j=1}^{f} \boldsymbol{W}_{(l)}[j,i] \cdot \boldsymbol{P}_{(l)}[:,j]\Big), \qquad (8)$$

where $\boldsymbol{H}_{(l+1)}[:,i]$ denotes the $i$-th column vector of all nodes in $\boldsymbol{H}_{(l+1)}$, and the weight entry $\boldsymbol{W}_{(l)}[j,i]$ symbolizes the neuron mapping the $j$-th embedding feature to the $i$-th representation feature. UNIFEWS sparsification on the weight matrix can thus be presented in the entry-wise manner following weight threshold $\delta_w$:

$$\hat{\boldsymbol{W}}[j,i] = \mathrm{thr}_{\delta_w'}\big(\boldsymbol{W}[j,i]\big) \cdot \boldsymbol{W}[j,i], \ \delta_w' = \delta_w/\|\boldsymbol{P}[:,j]\|. \quad (9)$$

**Approximation Bound.** To derive the precision bound for UNIFEWS on iterative models, we investigate the difference of layer representation:

$$\hat{\boldsymbol{H}}_{(l+1)} - \boldsymbol{H}_{(l+1)} = (\hat{\boldsymbol{P}}_{(l)} - \boldsymbol{P}_{(l)})\boldsymbol{W}_{(l)} + \hat{\boldsymbol{P}}_{(l)}(\hat{\boldsymbol{W}}_{(l)} - \boldsymbol{W}_{(l)}).$$

The first term is similar to graph propagation in Proposition 4.3, while the second term can be bounded by the weight pruning scheme regarding $\delta_w$. Hence, we are able to show that the representation under layer-progressive UNIFEWS for iterative networks containing both graph and weight operations satisfies the following proposition:

**Proposition 4.4** (**Progressive UNIFEWS for iterative update**). *For an L-round iterative update under Algorithm 2, the approximation error on output $\|\hat{\boldsymbol{H}}_{(L)} - \boldsymbol{H}_{(L)}\|_F$ is bounded by $O(\epsilon\|\boldsymbol{H}_{(L)}\|_F + \delta_w)$.*

The detailed interpretation of Proposition 4.4 is discussed in Appendix B.5. In brief, the approximation of layer representation is jointly bounded by factors depicting graph and

weight sparsification processes. Hence, the unified pruning produces an advantageous approximation of the learned representations across GNN layers, which completes our framework for characterizing the approximation bound of UNIFEWS for general GNN schemes by examining the graph smoothing optimization throughout model learning.

**Complexity Analysis.** As shown in Algorithms 1 and 2, entry-level operations can be naturally inserted into the computation *without* additional overhead. For graph propagation, when the graph sparsity is $\eta_a = q_a/m$, where $q_a$ is the number of removed edges, the computation complexity for propagation in each layer is reduced from $O(m)$ to $O\big((1 - \eta_a)m\big)$. In particular, for iterative models, this applies to both time and memory overhead. For weight pruning regarding matrices $\boldsymbol{P}, \boldsymbol{H} \in \mathbb{R}^{n \times f}$ and $\boldsymbol{W} \in \mathbb{R}^{f \times f}$, let the pruning ratio of weight matrix be $\eta_w$. The sparsification scheme at least reduces complexity to $O\big((1 - \eta_w)nf^2\big)$. The favorable merit of joint graph and weight sparsification by UNIFEWS is that, both the propagation result $\boldsymbol{P}$ and the weight multiplication product $\boldsymbol{H}$ enjoy sparsity from the previous input alternatively. In fact, as proven by (Zhang et al., 2023b), the scale of reduced computational operation can be advanced to $(1 - \eta_w)^2$ for certain distribution.

## 5. Experimental Evaluation

**Datasets and Metrics.** In the main experiment, we adopt 6 representative datasets including 3 small-scale (Kipf & Welling, 2017) and 3 large-scale ones (Hu et al., 2020) considering the applicability of evaluated methods. We mainly focus on the transductive node classification task. The efficiency is comprehensively assessed by computation time and floating-point operations (FLOPs), and 1FLOPs $\approx$ 2MACs. More datasets are evaluated in Appendix C.1.

**Backbone Models and Sparsification Methods.** We employ GCN (Kipf & Welling, 2017) and GAT (Brody et al., 2022) as *iterative* backbone GNNs, since they are commonly used for sparsification. Four graph and joint compression methods are considered as baselines: GLT (Chen et al., 2021), GEBT (You et al., 2022), CGP (Liu et al., 2023a), and DSpar (Liu et al., 2023b). Additionally, we examine the random entry-wise scheme RAN by randomly removing entries according to the given sparsity. Backbones of *decoupled* models are SGC (Wu et al., 2019) and APPNP (Klicpera et al., 2019). Two baselines with propagation personalization techniques, NDLS (Zhang et al., 2021) and NIGCN (Huang et al., 2023), are used, while both are only available for SGC.

**Hyperparameters.** We commonly utilize graph normalization $r = 0.5$, model layer depth $L = 2$, and layer width $f_{hidden} = 512$. For decoupled models, the number of propagation hops is 20. We employ full-batch and mini-batch training for iterative and decoupled methods, respectively. The total number of training epochs is 200, including pre-

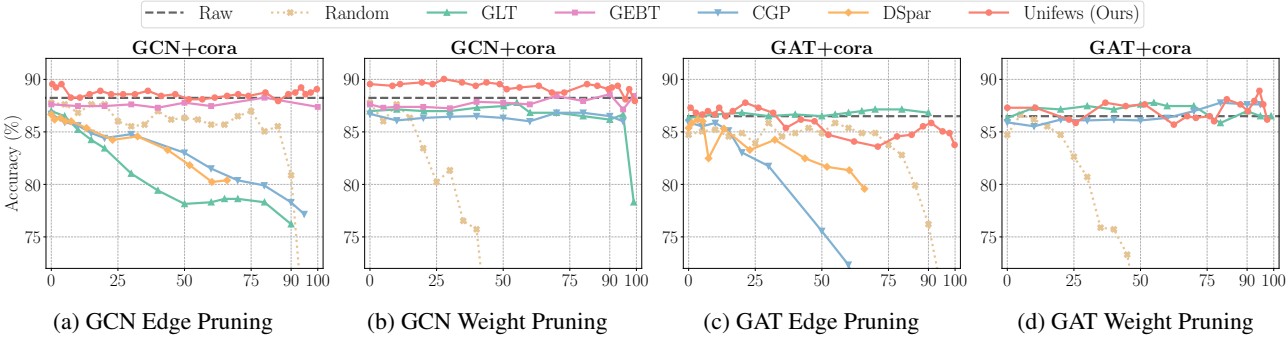

(a) GCN Edge Pruning     (b) GCN Weight Pruning     (c) GAT Edge Pruning     (d) GAT Weight Pruning

Figure 3: Accuracy of *iterative* models over cora, while more results are in Figure 10. Columns of "Edge" and "Weight Sparsity" present the average results of models with solely edge and weight sparsification, respectively.

training or fine-tuning process in applicable methods. The batch size is 512 for small datasets and 16384 for large ones. We tune the edge and weight sparsity of evaluated models across their entire available ranges.

### 5.1. Performance Comparison

We first separately apply only one part of sparsification, i.e., either edge or weight pruning, for better comparison. Figure 3 presents accuracy results of iterative backbones and compression methods over representative datasets. GEBT encounters out of memory error on pubmed. For larger graphs, most of the iterative baselines suffer from the out of memory error due to the expense of trainable adjacency matrix design and full-graph training process.

**Edge Sparsification.** For the *iterative* architecture in Figures 3(a) and 3(c), UNIFEWS outperforms state-of-the-art graph and joint compression approaches in most backbone-dataset combinations. Typically, for relatively small ratios $\eta_a < 80\%$, models with UNIFEWS pruning achieve comparable or exceeding accuracy, aligning with our approximation analysis. For higher sparsity, UNIFEWS benefits from the skip connection design, which carries essential information of node identity and therefore retains accuracy no worse than the trivial transformation. On the contrary, most of the competitors experience significant accuracy drop on one or more datasets. CGP and DSpar exhibit poor utility on the GAT backbone since its variable connections are more vulnerable to removal. Additionally, the comparison with random sparsification indicates that, the entry-wise scheme is particularly effective for small ratios, as the randomized pruning even surpasses dedicated methods for some circumstances. For high edge sparsity, the UNIFEWS preservation of dominant edges and identity mapping is critical to accuracy, which validates the advantage of our design.

Similarly, for *decoupled* propagation in Figure 4, UNIFEWS is able to preserve remarkable accuracy and outperform personalized propagation methods. On citeseer, it raises SGC accuracy by 2% through mitigating the over-smoothing issue. We hence conclude that, compared to heuristic pruning schemes, UNIFEWS successfully removes unimportant

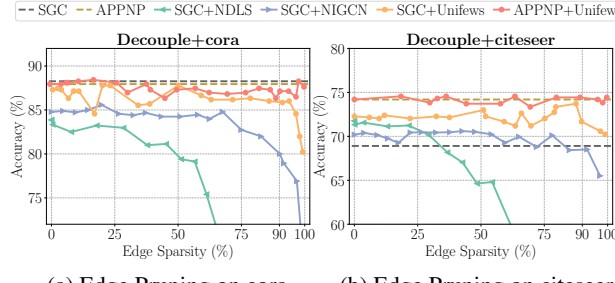

(a) Edge Pruning on cora     (b) Edge Pruning on citeseer

Figure 4: Accuracy of *decoupled* models over cora and citeseer. UNIFEWS is employed with solely edge removal.

graph edges without sacrificing efficacy, thanks to its fine-grained and adaptive scheme.

**Weight Sparsification.** Figures 3(b) and 3(d) display the efficacy of network pruning. For all combinations of models and graphs, UNIFEWS achieves top-tier accuracy with a wide range of weight sparsity. For small ratios, the model produces up to 3% accuracy gain over the bare GNN, resembling the benefit of neural network compression (Hoefler et al., 2021). Most evaluated baselines also maintain low error rate compared to their performance in graph pruning, indicating that the GNN weights are relatively redundant and are suitable for substantial compression. Thanks to the redundancy in GNN architecture, baselines including GEBT and CGP present strong performance in certain cases compared to edge sparsification. Random removal fails for high weight sparsity, which implies that the magnitude-based scheme is the key to maintain model effectiveness.

**Joint Sparsification.** For unified pruning on graph edges and network weights, accuracy comparison is provided by Figure 5 with applicable methods. It is noticeable that UNIFEWS retains comparable or better accuracy than the backbone GCN with up to 3% improvement. Most baseline methods only obtain suboptimal accuracy especially for weight pruning, which is affected by their comparatively poor graph sparsification. While GEBT mostly retains effectiveness on cora, it is highly limited by the specific architecture and high training overhead. The evaluation further

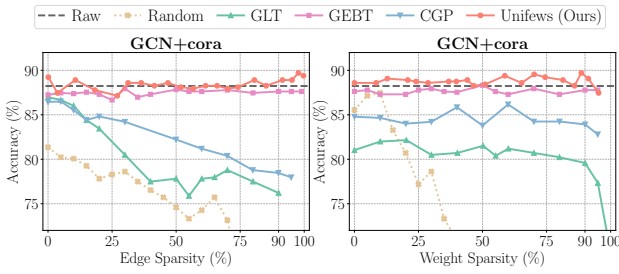

(a) Edge Pruning, $\eta_w = 30\%$ (b) Weight Pruning, $\eta_a = 30\%$

Figure 5: Accuracy of *joint sparsification* with varying edge and weight ratios, while fixing the other at $30\%$.

highlights the advantage of UNIFEWS in considering the two GNN operation stages in a unified manner.

### 5.2. Efficiency Analysis

**FLOPs.** As graph computation is the primary bottleneck for large-scale tasks, we particularly study the efficiency improvement utilizing decoupled models in Table 1. Evaluation implies that, UNIFEWS is effective in producing operational reduction proportional to the sparsity, i.e., saving $50\%$ computation FLOPs. It also achieves higher compression such as for APPNP over products, by recognizing more unnecessary edges than required. Contrarily, NDLS suffers from out of memory error on large datasets due to its complex calculation and implementation, while NIGCN does not guarantee decreased computation because of its expansionary propagation design. Table 1 also presents the transformation FLOPs for reference, where the sparsified embedding of UNIFEWS also benefits the downstream computation. We remark that, although the FLOPs value for transformation stage appears to be on the same level with propagation, in practice it can be efficiently processed utilizing parallel computation on devices such as GPU (Liu et al., 2020; Peng et al., 2022). Hence the GNN scalability bottleneck, especially on large datasets, is still the graph propagation. In this context, the ability of UNIFEWS in

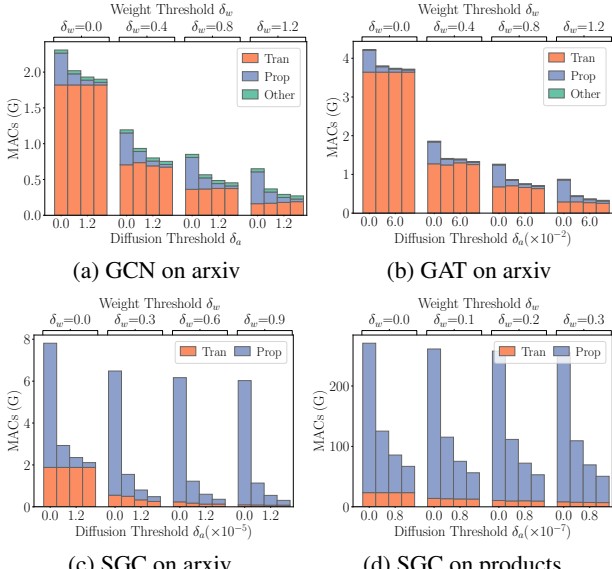

(a) GCN on arxiv (b) GAT on arxiv

(c) SGC on arxiv (d) SGC on products

Figure 6: Propagation and transformation FLOPs. Iterative models: **(a)** GCN and **(b)** GAT over arxiv. Decoupled models: SGC over **(c)** arxiv and **(d)** products.

reducing propagation operations is of unique importance.

**Runtime.** Within each method in Table 1, the propagation time is correlated with its FLOPs, and escalates rapidly with the data size due to the nature of graph operations. Comparison across methods demonstrates that UNIFEWS excels in efficiently conducting graph propagation with regard to both FLOPs and execution time. It realizes significant acceleration on large graphs by favorably reducing the graph scale across all layers, which benefits the system workload such as better entry-level access. The papers100m result highlights the superiority of UNIFEWS with $85 - 100\times$ improvement over the backbone and $9\times$ speed-up over NIGCN.

**Operation Breakdown.** To specifically evaluate the efficiency enhancement, in Figure 6, we separately assess the FLOPs related to graph propagation and network transformation for different backbone models and datasets under

Table 1: Results of $\eta_a = 50\%$ graph sparsification on decoupled models. "Time" denotes propagation time (in seconds), while "FLOPs" separately presents the operations (in GMACs) for propagation and transformation on all nodes. "Improvement" is the comparison between UNIFEWS and the backbone model, where improved accuracy is marked in **bold** fonts.

| Dataset | cora | | | citeseer | | | pubmed | | | arxiv | | | products | | | papers100m | | |
|---|---|---|---|---|---|---|---|---|---|---|---|---|---|---|---|---|---|---|
| Nodes $n$ | 2,485 | | | 3,327 | | | 19,717 | | | 169,343 | | | 2,400,608 | | | 111,059,956 | | |
| Edges $m$ | 12,623 | | | 9,228 | | | 88,648 | | | 2,315,598 | | | 123,718,024 | | | 3,228,124,712 | | |
| Metric | Acc | Time | FLOPs | Acc | Time | FLOPs | Acc | Time | FLOPs | Acc | Time | FLOPs | Acc | Time | FLOPs | Acc | Time | FLOPs |
| SGC | 85.8 | 0.13 | 0.36+2.2 | 67.7 | 0.44 | 0.68+2.8 | 83.0 | 0.36 | 0.89+2.3 | 68.8 | 3.9 | 5.9+15.0 | 79.1 | 289.6 | 247.4+434.4 | 63.3 | 19212 | 17.7+253.6 |
| +NDLS | 80.3 | 1362 | 0.19+2.7 | 64.7 | 1940 | 0.33+4.3 | 77.0 | 4717 | 0.42+2.0 | | (OOM) | | | (OOM) | | | (OOM) | |
| +NIGCN | 84.2 | 0.45 | 0.22+3.0 | 70.4 | 0.47 | 0.28+2.1 | 85.0 | 9.2 | 0.44+2.0 | 63.7 | 87.6 | 15.6+14.7 | 77.9 | 1026 | 137.2+182.0 | 53.7 | 1770 | 110.7+238.6 |
| **+UNIFEWS** | **86.0** | 0.10 | 0.18+1.2 | **73.0** | 0.26 | 0.35+1.7 | 83.0 | 0.24 | 0.47+2.0 | **69.4** | 1.5 | 3.0+13.3 | 78.5 | 203.1 | 124.0+186.9 | 63.1 | 192.4 | 5.3+143.8 |
| *Improvement* | 0.2 | 1.3× | 2.0×, 1.9× | 5.3 | 1.7× | 2.0×, 1.7× | 0.0 | 1.5× | 1.9×, 1.2× | 0.6 | 2.6× | 2.0×, 1.1× | -0.5 | 1.4× | 2.0×, 2.3× | -0.2 | 99.8× | 3.3×, 1.8× |
| APPNP | 86.2 | 0.15 | 0.36+2.2 | 71.6 | 0.43 | 0.68+2.8 | 87.6 | 0.33 | 0.89+2.3 | 64.8 | 2.6 | 5.9+20.9 | 72.5 | 248.5 | 247.4+269.4 | 60.9 | 15305 | 17.7+247.7 |
| **+UNIFEWS** | **86.5** | 0.08 | 0.18+1.8 | **73.7** | 0.21 | 0.31+2.3 | **88.0** | 0.26 | 0.43+1.8 | **65.0** | 0.93 | 3.0+15.0 | **76.9** | 58.5 | 31.7+186.9 | **62.8** | 178.5 | 8.9+241.8 |
| *Improvement* | 0.4 | 1.8× | 2.0×, 1.2× | 2.1 | 2.0× | 2.2×, 1.2× | 0.4 | 1.3× | 2.1×, 1.3× | 0.2 | 2.8× | 1.9×, 1.4× | 4.5 | 4.2× | 7.8×, 1.4× | 1.9 | 85.7× | 2.0×, 1.0× |

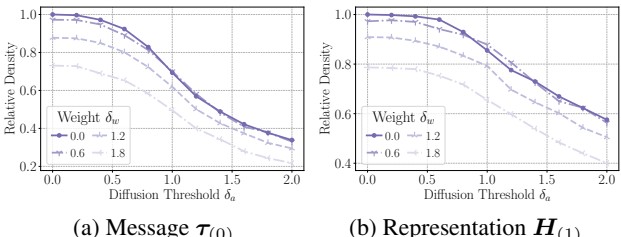

(a) Message $\boldsymbol{\tau}_{(0)}$      (b) Representation $\boldsymbol{H}_{(1)}$

Figure 7: Entry sparsity under joint sparsification, evaluated by matrix norm relative to the unpruned one on a 2-layer GCN over cora. **(a)** Edge-wise message $\|\hat{\boldsymbol{\tau}}_{(0)}\|/\|\boldsymbol{\tau}_{(0)}\|$. **(b)** Node-wise representation $\|\hat{\boldsymbol{H}}_{(1)}\|_F/\|\boldsymbol{H}_{(1)}\|_F$.

UNIFEWS sparsification. For better presentation, model width is set to 64 in this experiment. Figures 6(a) and 6(b) imply that the majority of computational overhead of *iterative models* is network transformation, even on graphs as large as arxiv. Consequently, weight compression is essential for reducing GNN operations.

On the other hand, graph propagation becomes the bottleneck in *decoupled designs*, and is increasingly significant on graphs with greater scales. This is because of the larger number of propagation hops $L = 20$ for these structures. In this case, UNIFEWS is effective in saving computational cost by simplifying the propagation and bypassing unnecessary operations, with over $20\times$ and $5\times$ joint reduction on arxiv and products, respectively. We also discover the benefit of joint pruning that a higher threshold results in smaller FLOPs of both operations in Figures 6(c) and 6(d), which signifies the win-win situation brought by increased sparsity. By combining these two sparsifications, we summarize that the unified scheme of UNIFEWS is capable of mitigating the computational overhead for both propagation- and transformation-heavy tasks.

### 5.3. Parameter Discussion

**Sparsity.** Figure 7 displays the relative density of the edge message and node representation matrices, respectively. It can be observed that the entry sparsity is enhanced with the increase of both two pruning ratios, signifying our theoretical analysis that UNIFEWS not only directly shrinks edges and weights, but promotes the sparsity of the product matrix as well. It also supports our claim that UNIFEWS is superior in employing dual sparsification where the two stages benefit each other alternatively and enjoy a win-win situation brought by the increased sparsity.

**Joint Sparsification Thresholds.** The thresholds controlling diffusion and weight sparsification are pivotal to UNIFEWS compression, affecting both efficacy and efficiency of the produced model. Figure 8 presents the changes of inference accuracy and dual sparsity of GNN models under graph and joint sparsification with varying adjacency thresholds $\delta_a$. Accuracy in the plot follows the conclusion in previous evaluation, that it only degrades above extreme

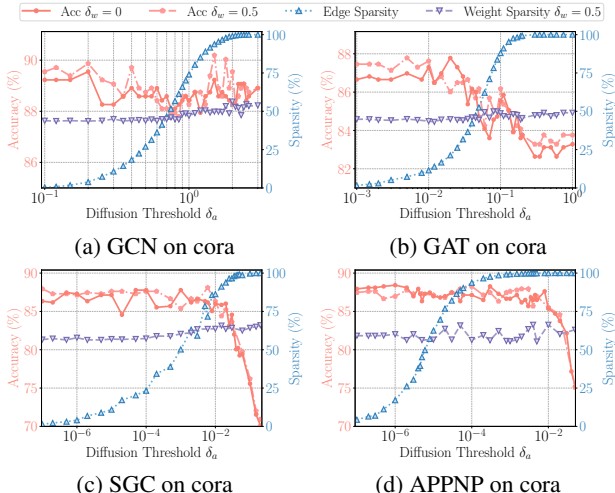

(a) GCN on cora      (b) GAT on cora

(c) SGC on cora      (d) APPNP on cora

Figure 8: Sensitivity of joint sparsification thresholds on four models over cora. We respectively set the weight ratio $\delta_w$ to $0$ and $0.5$, then evaluate the accuracy, edge sparsity, and weight sparsity of the model. Note that the x-axis representing diffusion threshold is on a logarithmic scale.

sparsity $\eta_a > 95\%$. The experiment reveals that, the relation between edge sparsity and adjacency threshold aligns with Theorem 4.2, and decoupled models typically require a larger range of threshold to traverse the sparsity range, which is because of the wider distribution of their entry values throughout the deeper propagation. Interestingly, the weight sparsity when $\delta_w = 0.5$ also increases under high edge pruning ratios. The pattern is also observed in the reverse case. We deduce that the reciprocal enhancement in graph and model sparsity is brought by the unified sparsification of UNIFEWS, which conforms to our analysis as well as previous studies (Zhang et al., 2023b).

## 6. Conclusion

In this work, we present UNIFEWS, an entry-wise GNN sparsification with a unified framework for graph edges and model weights. By bridging spectral graph smoothing and GNN sparsification, we showcase in theory that the layer-progressive UNIFEWS provides an effective approximation on the graph learning process with a close optimization objective, which is favorable for multi-layer GNN updates in both iterative and decoupled architectures. Comprehensive experiments underscore the superiority of UNIFEWS in terms of efficacy and efficiency, including comparable or improved accuracy, $90 - 95\%$ operational reduction, and up to $100\times$ faster computation.

**Limitation.** One potential limitation lies in the consideration of graph heterophily, as messages with large magnitudes may not be beneficial to model prediction. In this case, the graph smoothing objective in Definition 3.1 needs to be refined; consequently, the sparsification strategy should be adjusted.

## Acknowledgements

This research is supported by Singapore MOE AcRF Tier-2 Grant (T2EP20122-0003) and NTU-NAP startup grant (022029-00001). Ningyi Liao is supported by the Joint NTU-WeBank Research Centre on FinTech, Nanyang Technological University, Singapore.

## Impact Statement

This paper presents work whose goal is to advance the field of Machine Learning. There are many potential societal consequences of our work, none which we feel must be specifically highlighted here.

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

# A. Detailed Related Works and Comparison

## A.1. GNNs Simplification

**Iterative Graph Sparsification.** Simplification techniques mainly aim to reduce GNN message-passing operations while maintaining its scheme. Consequently, these methods usually enjoy better expressiveness and generality to different model architectures. Among them, a large portion of works sparsify edge connections for accuracy improvement, typically through deleting edges. NeuralSparse (Zheng et al., 2020) and LSP (Kosman et al., 2022) drop edges based on $k$-neighbor metric for each node and layer-wise local graph topology, respectively, while AdaptiveGCN (Li et al., 2021) and SGCN (Li et al., 2022) choose to implement sparsity by learnable edge features and graph adjacency.

In comparison, FastGAT (Srinivasa et al., 2020) and DSpar (Liu et al., 2023b) relate graph modification to *spectral sparsification* mainly as an attempt to boost efficiency. FastGAT eliminates connections in the GAT (Veličković et al., 2018) model by calculating the Effective Resistance (Spielman & Srivastava, 2011) but at the expense of high computational time. DSpar employs a loose approximation and successfully applies to larger datasets.

Most of the simplification techniques, however, conduct graph modification in a one-time and layer-agnostic manner, resulting in limited flexibility and utility under the trade-off between efficiency and efficacy.

**Decoupled Propagation Personalization.** To make use of the model propagation and perform more adaptive maneuvers, some studies design fine-grained optimizations on each diffusion step in a node-dependent fashion to resolve GNN defects such as inefficiency (Chiang et al., 2019; Chen et al., 2020; Liu et al., 2025; Liao et al., 2026) and over-smoothing (Li et al., 2018; Liao et al., 2025). NDLS (Zhang et al., 2021) and NIGCN (Huang et al., 2023) personalize the hop number per node in the decoupled GNN design as an approach to acknowledge the local structure and mitigate over-smoothing. SCARA (Liao et al., 2022; 2023b) and LD$^2$ (Liao et al., 2023a) replace the hop-base message-passing with a graph-oriented pushing scheme and eager termination, while Shadow-GNN (Zeng et al., 2021) explicitly selects the appropriate node-wise aggregation neighborhood.

Another set of works (Lai et al., 2020; Spinelli et al., 2021; Miao et al., 2021; Zhang et al., 2022) achieves customized diffusion by gradually learning the hop-level importance of nodes at the price of introducing more overhead, which are less relevant to the efficient GNN computation.

**Architecture Compression.** The concept of GNN pruning takes the neural network architecture into account and exploits sparsification for efficiency without hindering effectiveness (Zhang et al., 2023b). (Zhou et al., 2021) consider structural channel pruning by reducing the dimension of weight matrices, while CGP (Liu et al., 2023a) appends a prune-and-regrow

Table 2: Summary of primary symbols and notations.

| Notation | Description |
| --- | --- |
| $\mathcal{G}, \mathcal{V}, \mathcal{E}$ | Graph, node set, and edge set |
| $\mathcal{N}(u)$ | Neighboring node set of node $u$ |
| $n, m, f$ | Node, edge, and feature size |
| $L$ | Number of propagation hops and network layers |
| $c$ | Graph smoothing regularization coefficient |
| $b$ | Graph smoothing gradient descent step size |
| $\epsilon$ | Graph spectral approximation rate |
| $\delta_a, \delta_w$ | Thresholds for graph and weight sparsification |
| $q_a, q_w$ | Numbers of removed edges and weight entries |
| $\eta_a, \eta_w$ | Graph sparsity and weight sparsity |
| $\bar{A}, \tilde{A}$ | Self-looped and normalized adjacency matrix of graph $\mathcal{G}$ |
| $\bar{L}, \tilde{L}$ | Raw and normalized Laplacian matrix of graph $\mathcal{G}$ |
| $\hat{A}, \hat{L}$ | Approximate adjacency and Laplacian matrix of graph $\hat{\mathcal{G}}$ |
| $T, W$ | Graph diffusion matrix and weight matrix |
| $\tau[u, v]$ | Message entry corresponding to edge $(u, v)$ |
| $\omega[j, i]$ | Network entry corresponding to neuron $(j, i)$ |
| $X, x$ | Node attribute matrix and feature-wise vector |
| $P_{(l)}, p_{(l)}$ | Embedding matrix and feature-wise vector of layer $l$ |
| $H_{(l)}, h_{(l)}$ | Representation matrix and feature-wise vector of layer $l$ |

scheme on the graph structure.

A line of research refers to the *lottery ticket hypothesis* (Frankle & Carbin, 2019; Hui et al., 2023) in the context of graph learning in search of a smaller yet effective substructure in the network. Among these approaches, GEBT (You et al., 2022) finds the initial subnetwork with simple magnitude-based pruning on both adjacency and weight matrices, whilst ICPG (Sui et al., 2024) learns the importance of edges as an external task. GLT (Chen et al., 2021) and DGLT (Wang et al., 2023b) employ an alternative optimization strategy that progressively processes the graph and network components. Other compression techniques such as *quantization* (Ding et al., 2021; Bahri et al., 2021; Wang et al., 2021b) are also investigated.

We mark the difference between our work and compression methods that the latter graph and network sparsification are loosely coupled, usually entailing a training procedure on the full-size graph and model to identify sparsification components, which prevents their application to large-scale data.

### A.2. GNNs Modification

**Graph Sampling.** Different from simplification, modification techniques change the message-passing scheme in GNNs, leading to new models with distinct expressiveness after modification. Sampling is a common strategy for improving GNN efficiency by altering the message-passing process. It is based on the intuition that the learning process only require a portion of data samples for each iteration. Thence, strategies on various hierarchies are designed for retrieving useful information (Chen et al., 2018; Huang et al., 2018; Zou et al., 2019; Zeng et al., 2020; Rong et al., 2020; Fang et al., 2023). A contemporary work (Ding et al., 2025) also investigates the utilization of spectral sparsification for sampling GNNs in the polynomial form, which echos with our theoretical analysis that graph sparsification process is useful in offering bounded approximation for message-passing GNNs. We however note that the sampling approach exhibits iterative processing of the graph components and does not produce a deterministic sparsified graph. Consequently, its capability on decoupled models, which are dominant on large graphs, is relatively constrained.

**Graph Coarsening.** Graph coarsening describes another modification approach that reduces the graph size by contracting nodes into subsets. GraphZoom (Deng et al., 2020b), GOREN (Cai et al., 2021), and (Huang et al., 2021) explore various coarsening schemes and enhance training scalability, while GCond (Jin et al., 2022) alternatively adopts an analogous condensation strategy.

### A.3. Comparative Discussion

We highlight three advantages of our UNIFEWS approach compared with above simplification and modification techniques in the context of GNN learning. Firstly, realistic graph propagation and feature transformation are conducted in an entry-centric fashion. Therefore, entry-level operations can be naturally inserted into the computation *on the fly* by modifying the matrix multiplication operator. Secondly, different from previous node- or edge-dependent strategies, our sparsification operates on the messages being passed during propagation, which is inherently informative to foster effective pruning while retaining model *expressivity*. Lastly, the *progressive* pruning across layers is capable of promoting greater graph sparsity, thereby alleviating intrinsic drawbacks of the GNN architecture including over-smoothing and neighborhood explosion. We empirically showcase the improvement on entry sparsity in Appendix C.3.

## B. Detailed Proof and Theoretical Analysis

### B.1. Proof of Lemma 3.3

*Proof.* By using the closed-form solution $p^* = (I + cL)^{-1}x$ and the fact that $A^{-1} - B^{-1} = B^{-1}(B - A)A^{-1}$, we have:

$$\hat{p}^* - p^* = \left((I + c\hat{L})^{-1} - (I + cL)^{-1}\right)x$$

$$= (I + c\hat{L})^{-1}(c\hat{L} - cL)(I + cL)^{-1}x = c(I + c\hat{L})^{-1}(\hat{L} - L)p^*.$$

From Equation (2), we can acquire the difference between the raw and approximate Laplacian matrices based on the spectral property:

$$\|L - \hat{L}\|_2 = \sup_{\|x\|=1} x^\top(L - \hat{L})x = x_0^\top(L - \hat{L})x_0 \leq \epsilon x_0^\top I x_0 = \epsilon, \tag{10}$$

where $\|\cdot\|_2$ is the matrix spectral norm, and the supremum is achieved when $x = x_0$.

The distance between $p^*$ and $\hat{p}^*$ follows the consistency of spectral norm $\|Ax\| \leq \|A\|_2 \|x\|$. By substituting Equation (10)

and utilizing the property of spectral norm, we have:

$$\|\hat{\boldsymbol{p}}^* - \boldsymbol{p}^*\| \leq c\|(\boldsymbol{I} + c\hat{\boldsymbol{L}})^{-1}\|_2 \cdot \|\hat{\boldsymbol{L}} - \boldsymbol{L}\|_2 \cdot \|\boldsymbol{p}^*\|$$

$$= c\epsilon\|\boldsymbol{p}^*\| \cdot \max_i \left\{ \frac{1}{\lambda_i(\boldsymbol{I} + c\hat{\boldsymbol{L}})} \right\} = \frac{c\epsilon\|\boldsymbol{p}^*\|}{1 + c\lambda_1(\hat{\boldsymbol{L}})} = c\epsilon\|\boldsymbol{p}^*\|,$$

where $\lambda_i(\boldsymbol{A})$ denotes the $i$-th smallest eigenvalue of matrix $\boldsymbol{A}$. $\qquad\square$

Given the additive nature of the graph specifier, the bound for graph smoothing problem Lemma 3.3 is dissimilar to the approximation setting with multiplicative similarity bounded by the quadratic form $O(c\epsilon\boldsymbol{p}^\top \boldsymbol{L}\boldsymbol{p})$ (Sadhanala et al., 2016; Calandriello et al., 2018), but instead correlates with the embedding vector norm $\|\boldsymbol{p}^*\|$. This correlation arises from the bias introduced by the pruned entries in the diffusion matrix, which is associated with the embedding value.

### B.2. Proof of Lemma 4.1

*Proof.* We outline the diffusion by the general graph adjacency $\boldsymbol{T} = \boldsymbol{A}$. The entry-wise difference matrix for the sparsified diffusion can be derived as $\boldsymbol{\Upsilon} = \boldsymbol{A} - \hat{\boldsymbol{A}} = \hat{\boldsymbol{L}} - \boldsymbol{L}$. Additionally, the pruned edges form the complement set $\mathcal{E}_\Upsilon = \mathcal{E} \setminus \hat{\mathcal{E}}$, and the number of removed edges is $q_a = |\mathcal{E}_\Upsilon|$. If an edge is pruned $(u, v) \in \mathcal{E}_\Upsilon$, the entry $\boldsymbol{\Upsilon}[u, v] = \boldsymbol{A}[u, v]$.

For a current embedding $\boldsymbol{p}$, its product with the difference matrix $\boldsymbol{\Upsilon}\boldsymbol{p}$ only correlates with entries that have been pruned. Based on the Minkowski inequality and sparsification scheme Equation (5), the $L_1$ norm of the product vector satisfies:

$$\|\boldsymbol{\Upsilon}\boldsymbol{p}\|_1 = \sum_{u\in\mathcal{V}} \left| \sum_{v\in\mathcal{N}_\Upsilon(u)} \boldsymbol{A}[u,v]\boldsymbol{p}[v] \right| = \sum_{u\in\mathcal{V}} \left| \sum_{v\in\mathcal{N}_\Upsilon(u)} \tau[u,v] \right|$$

$$\leq \sum_{u\in\mathcal{V}} \sum_{v\in\mathcal{N}_\Upsilon(u)} \left| \tau[u,v] \right| \leq \sum_{u\in\mathcal{V}} \sum_{v\in\mathcal{N}_\Upsilon(u)} \delta_a = q_a\delta_a.$$

Employing the relationship between vector norms given by the Cauchy-Schwarz inequality $\|\boldsymbol{x}\| \leq \|\boldsymbol{x}\|_1$, the difference of quadratic forms with regard to graph Laplacian can be bounded as:

$$|\boldsymbol{p}^\top \boldsymbol{L}\boldsymbol{p} - \boldsymbol{p}^\top \hat{\boldsymbol{L}}\boldsymbol{p}| = |\boldsymbol{p}^\top \boldsymbol{\Upsilon}\boldsymbol{p}| \leq \|\boldsymbol{p}\| \cdot \|\boldsymbol{\Upsilon}\boldsymbol{p}\| \leq q_a\delta_a\|\boldsymbol{p}\|.$$

Referring to Definition 3.2, Equation (2) is equivalent to:

$$\left| \boldsymbol{x}^\top (\boldsymbol{L} - \hat{\boldsymbol{L}})\boldsymbol{x} \right| \leq \epsilon \cdot \|\boldsymbol{x}\|^2, \quad \forall \boldsymbol{x} \in \mathbb{R}^n. \tag{11}$$

Therefore, when the condition $q_a\delta_a \leq \epsilon\|\boldsymbol{p}\|$ is met, the sparsified $\hat{\boldsymbol{L}}$ is spectrally similar to $\boldsymbol{L}$ with approximation rate $\epsilon$. $\quad\square$

### B.3. Proof of Theorem 4.2

*Proof.* Recall in Equation (5) pruning, there is $\mathcal{E}_\Upsilon = \{(u, v) \mid |\tau[u, v]| \leq \delta_a\}$. The fraction of entries below the threshold is thence expressed by the probability:

$$q_a = |\mathcal{E}_\Upsilon| = m \cdot P(\mathbf{A}[u, v] \cdot |\mathbf{p}[v]| \leq \delta_a)$$

$$= 2m \int_0^\infty P(\mathbf{A}[u, v] \leq \frac{\delta_a}{x}) \cdot f_{p[v]}(x)dx,$$

where $f_{p[v]}(x)$ represents the probability distribution function of embedding values. As $\mathbf{A}[u, v] \geq x_{min}$, we have $P(\mathbf{A}[u, v] \leq \frac{\delta_a}{x}) = 0$, when $x > \frac{\delta_a}{x_{min}}$. The integral upper bound changes to $\frac{\delta_a}{x_{min}}$, i.e.

$$q_a = 2m \int_0^{\frac{\delta_a}{x_{min}}} P(\mathbf{A}[u, v] \leq \frac{\delta_a}{x}) \cdot f_{p[v]}(x)dx.$$

Acknowledging the assumptions, in scale-free graphs, the distribution of nodes follows the power law $P(d) = d^{-\alpha}$, where $P(d)$ is the fraction of nodes of degree $d$ for large values, and $\alpha$ is a constant typically within range $2 < \alpha < 3$. For the normalized adjacency matrix $\tilde{\boldsymbol{A}} = \boldsymbol{D}^{-1/2}\bar{\boldsymbol{A}}\boldsymbol{D}^{-1/2}$, its entry distribution also follows the power law as $x^{-\alpha}$. The

embedding value $p[v] \sim N(0, \sigma^2)$. By substituting the distributions we have:

$$q_a = 2m \int_0^{\frac{\delta_a}{x_{min}}} \left[ 1 - \left( \frac{\delta_a}{x x_{min}} \right)^{-\alpha+1} \right] \frac{1}{\sqrt{2\pi\sigma^2}} \exp\left( -\frac{x^2}{2\sigma^2} \right) dx$$

$$= 2m \left[ \frac{1}{2} \operatorname{erf}\left( \frac{\delta_a/x_{\min}}{\sqrt{2}\,\sigma} \right) - \int_0^{\frac{\delta_a}{x_{min}}} \left( \frac{\delta_a}{x x_{min}} \right)^{-\alpha+1} \frac{1}{\sqrt{2\pi}\sigma} \exp\left( -\frac{x^2}{2\sigma^2} \right) dx \right]$$

$$= 2m\left( \frac{1}{2} - \frac{1}{\sqrt{2\pi}\sigma} \left( \frac{\delta_a}{x_{min}} \right)^{-\alpha+1} \int_0^{\frac{\delta_a}{x_{min}}} x^{\alpha-1} \exp\left( -\frac{x^2}{2\sigma^2} \right) dx \right),$$

$$\text{where} \quad \operatorname{erf}(z) = \frac{2}{\sqrt{\pi}} \int_0^z e^{-t^2} dt, \quad \gamma(s, x) = \int_0^x t^{s-1} e^{-t} dt.$$

The last term can be given by the lower incomplete Gamma function:

$$\int_0^{\frac{\delta_a}{x_{min}}} x^{\alpha-1} \exp\left( -\frac{x^2}{2\sigma^2} \right) dx = \sigma^\alpha \, 2^{\frac{\alpha-2}{2}} \, \gamma\left( \frac{\alpha}{2}, \frac{(\delta_a/x_{\min})^2}{2\sigma^2} \right).$$

Therefore,

$$q_a = m\left[ \operatorname{erf}\left( \frac{\delta_a/x_{\min}}{\sqrt{2}\,\sigma} \right) - \left( \frac{x_{\min}}{\delta_a} \right)^{\alpha-1} \cdot \frac{\sigma^\alpha \, 2^{\frac{\alpha}{2}}}{\sqrt{2\pi}\,\sigma} \, \gamma\left( \frac{\alpha}{2}, \frac{(\frac{\delta_a}{x_{\min}})^2}{2\,\sigma^2} \right) \right]. \tag{12}$$

To understand Equation (12), we discuss it in two extreme cases.

**Case 1.** When $\delta_a \to 0$, only a small number of edges are affected. Considering the Taylor Expansion with respect to $\delta_a$, there is:

$$q_a = c_1 \, \delta + c_3 \, \delta^3 + \cdots,$$

where $c_1 = \frac{m\sqrt{2/\pi}}{x_{\min}\sigma} \frac{\alpha-1}{\alpha}$.

**Case 2.** When $\delta_a > 0$ is sufficiently large, this is the main case we concerned where a portion of edges are pruned. The formula will be approximated by:

$$q_a = m - 2m \frac{2^{\frac{\alpha-3}{2}} \Gamma\left( \frac{\alpha}{2} \right)}{\sqrt{\pi}\sigma^{\alpha+1}} \left( \frac{\delta_a}{x_{min}} \right)^{1-\alpha},$$

or equivalently, $\eta_a = q_a/m = 1 - C\delta_a^{1-\alpha}$. Therefore, the relative strength of sparsification represented by the threshold $\delta_a/\|\boldsymbol{p}\|$ and the relative sparsity $\eta_a$ can be represented by each other.

Referring to Lemma 4.1, the approximation bound is given as:

$$\epsilon = O\left( \eta_a (1 - \eta_a)^{\frac{1}{1-\alpha}} \right), \tag{13}$$

which corresponds to Theorem 4.2 for giving the approximation bound. $\square$

### B.4. Proof Sketch of Proposition 4.3

Consider consecutively sparsifying the diffusion for each hop by Equation (7). Intuitively, as the edges are gradually removed from the original graph, there is $q_{a,(l_1)} < q_{a,(l_2)}$ for $l_1 < l_2$ with the same relative threshold. If the sparsest graph $\boldsymbol{T}_{(L)}$ satisfies Definition 3.1, then the multi-layer update is also bounded.

Recall that common GNN learning can be expressed by the graph smoothing process Definition 3.1. Such optimization problem can be iteratively solved by employing a gradient descent scheme (Ma et al., 2021), where each iteration derives the $l$-th hop of graph propagation as in Equation (1):

$$\boldsymbol{p}_{(l+1)} = \boldsymbol{p}_{(l)} - \frac{b}{2} \cdot \frac{\partial \mathcal{L}}{\partial \boldsymbol{p}} \bigg|_{\boldsymbol{p} = \boldsymbol{p}_{(l)}} = (1-b)\boldsymbol{p}_{(l)} - bc\boldsymbol{L}\boldsymbol{p}_{(l)} + b\boldsymbol{x}. \tag{14}$$

where $b/2$ is the step size and initially there is $\boldsymbol{p}_{(0)} = \boldsymbol{x}$. Equation (14) is expressive to represent various propagation operations in decoupled GNN models. For example, APPNP (Klicpera et al., 2019) can be achieved by letting $b = \alpha$ and $c = (1-\alpha)/\alpha$, while SGC (Wu et al., 2019) is the edge case with only the graph regularization term.

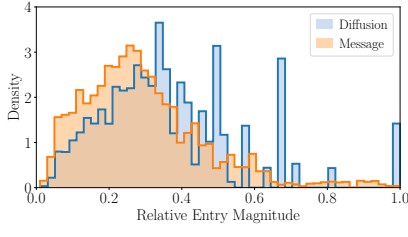

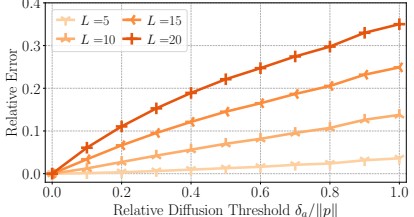

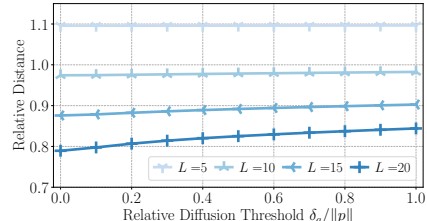

(a) Distribution of Entries

(b) Approximate Embedding Error

(c) Distance to Over-smoothed Embedding

Figure 9: Empirical evaluation on the SGC graph propagation and the effect of UNIFEWS on cora dataset. **(a)** Distribution of entries in the diffusion matrix $\boldsymbol{T}$ and the message $\boldsymbol{\tau}_{(0)}$ at hop $l = 0$. **(b)** The relative error margin $\|\hat{\boldsymbol{P}}_{(L)} - \boldsymbol{P}_{(L)}\|_F / \|\boldsymbol{P}_{(L)}\|_F$ of approximation to the raw embedding against the strength of UNIFEWS and propagation hops. **(c)** The relative distance $\|\hat{\boldsymbol{P}}_{(L)} - \boldsymbol{P}^*\|_F / \|\boldsymbol{P}^*\|_F$ to the converged embedding against the strength of UNIFEWS and propagation hops.

Now consider the layer-wise graph sparsification under Equation (14) updates. For the initial state, there is $\boldsymbol{p}_{(0)} = \hat{\boldsymbol{p}}_{(0)} = \boldsymbol{x}$. If in the $l$-th hop, Equation (5) edge sparsification is applied to the graph $\boldsymbol{L}$, then the approximation gap is:

$$\boldsymbol{p}_{(l+1)} - \hat{\boldsymbol{p}}_{(l+1)} = (1 - b)(\boldsymbol{p}_{(l)} - \hat{\boldsymbol{p}}_{(l)}) + bc(\hat{\boldsymbol{L}}\hat{\boldsymbol{p}}_{(l)} - \boldsymbol{L}\boldsymbol{p}_{(l)}). \tag{15}$$

To demonstrate that $\hat{\boldsymbol{p}}_{(l+1)}$ is an $\epsilon$-approximation, we use induction by assuming $\hat{\boldsymbol{p}}_{(l)} - \boldsymbol{p}_{(l)} = \boldsymbol{\Delta}_p, \|\boldsymbol{\Delta}_p / \boldsymbol{p}_{(l)}\| \sim O(\epsilon)$. Then the bound for approximation follows:

$$\|\boldsymbol{p}_{(l+1)} - \hat{\boldsymbol{p}}_{(l+1)}\| = \| - (1 - b)\boldsymbol{\Delta}_p + bc(\hat{\boldsymbol{L}}\boldsymbol{p}_{(l)} + \hat{\boldsymbol{L}}\boldsymbol{\Delta}_p - \boldsymbol{L}\boldsymbol{p}_{(l)})\|$$
$$\leq \|(bc\boldsymbol{L} - (1 - b)\boldsymbol{I})\|_2 \|\boldsymbol{\Delta}_p\| + bc\|\hat{\boldsymbol{L}} - \boldsymbol{L}\|_2 \|\boldsymbol{p}_{(l)} + \boldsymbol{\Delta}_p\|$$
$$\leq O(\epsilon) \cdot \|\boldsymbol{p}_{(l)}\| + bc\epsilon(1 + O(\epsilon))\|\boldsymbol{p}_{(l)}\|,$$

where the inequalities follows from the property of matrix spectral norm and Equation (10). Hence, the relative error of approximate representation $\hat{\boldsymbol{p}}_{(l+1)}$ is constrained by $O(\epsilon)$.

### B.5. Proof Sketch of Proposition 4.4

To apply the approximation analysis to iterative GNNs, we first extend the analysis to multi-feature input matrix. When there are $f$ input vectors, i.e., the input matrix is $\boldsymbol{X} \in \mathbb{R}^{n \times f}$, then the matrix form of graph Laplacian smoothing corresponding to Equation (1) is:

$$\boldsymbol{P}^* = \arg\min_{\boldsymbol{P}} \|\boldsymbol{P} - \boldsymbol{X}\|_F^2 + c \cdot \mathrm{tr}(\boldsymbol{P}^\top \boldsymbol{L} \boldsymbol{P}), \tag{16}$$

where $\|\cdot\|_F$ is the matrix Frobenius norm and the closed-form solution is $\boldsymbol{P}^* = (\boldsymbol{I} + c\boldsymbol{L})^{-1}\boldsymbol{X}$.

Since graph operations among feature dimensions are mutually independent, conclusion from Theorem 4.2 and Proposition 4.3 are still valid in their matrix forms. In iterative models, the gradient update similar to Equation (14) is instead employed to the representation matrix $\boldsymbol{H}_{(l)}$ and derives each layer as:

$$\boldsymbol{P}_{(l)} = (1 - b)\boldsymbol{H}_{(l)} - bc\boldsymbol{L}\boldsymbol{H}_{(l)} + b\boldsymbol{X}.$$

As detailed in (Ma et al., 2021; Zhu et al., 2021), the update scheme above is able to describe an array of iterative GNNs. For instance, when $\boldsymbol{H}_{(l)} = \boldsymbol{X}$ and $b = 1/c$, it yields the GCN propagation $\boldsymbol{P}_{(l)} = \tilde{\boldsymbol{A}}\boldsymbol{H}_{(l)}$. GAT convolution can be imitated by a node-wise $c$ depending on the attention values (Veličković et al., 2018; Brody et al., 2022).

To interpret the effect of UNIFEWS sparsification, we first investigate the entry-wise graph pruning in Algorithm 2 and its outcome, i.e., the embedding matrix $\hat{\boldsymbol{P}}_{(l)}$. For simplicity, we assume the sparsification is only employed upon the $(l + 1)$-th layer and $\hat{\boldsymbol{H}}_{(l)} = \boldsymbol{H}_{(l)}$. Invoking Equation (10) and the fact that $\|\boldsymbol{AB}\|_F \leq \|\boldsymbol{A}\|_2 \|\boldsymbol{B}\|_F$, the margin of approximate embeddings can be written as:

$$\|\hat{\boldsymbol{P}}_{(l)} - \boldsymbol{P}_{(l)}\|_F \leq bc\|\hat{\boldsymbol{L}} - \boldsymbol{L}\|_2 \|\boldsymbol{H}_{(l)}\|_F \leq bc\epsilon\|\boldsymbol{H}_{(l)}\|_F.$$

Then, consider the weight pruning Equation (9). The entry-wise error is a composition of joint embedding and weight approximation:

$$\hat{\boldsymbol{\omega}}_{(l+1)}[j, i] - \boldsymbol{\omega}_{(l+1)}[j, i] = \hat{\boldsymbol{W}}_{(l)}[j, i]\hat{\boldsymbol{P}}_{(l)}[:, j] - \boldsymbol{W}_{(l)}[j, i]\boldsymbol{P}_{(l)}[:, j].$$

Let $\boldsymbol{M}_{(l)} = \boldsymbol{W}_{(l)} - \hat{\boldsymbol{W}}_{(l)}$. Recalling the iterative update scheme Equation (8), the total difference on linear transformation

$\boldsymbol{H}' = \boldsymbol{P}_{(l)}\boldsymbol{W}_{(l)}$ is built up by:

$$\hat{\boldsymbol{H}}' - \boldsymbol{H}' = \sum_{i,j=1}^{f} (\hat{\boldsymbol{P}}_{(l)}[:,j] - \boldsymbol{P}_{(l)}[:,j])\boldsymbol{W}_{(l)}[j,i] + \hat{\boldsymbol{P}}_{(l)}[:,j]\boldsymbol{M}_{(l)}[j,i].$$

Its first term corresponds to the embedding approximation:

$$\boldsymbol{\Delta}_P \leq \|\hat{\boldsymbol{P}}_{(l)} - \boldsymbol{P}_{(l)}\|_F \, \|\boldsymbol{W}\|_F \leq bc\epsilon\|\boldsymbol{H}_{(l)}\|_F\|\boldsymbol{W}\|_F,$$

and the second term adheres to weight sparsification as per Equation (9):

$$\boldsymbol{\Delta}_W \leq \sum_{i=1}^{f}\sum_{j=1}^{f}\left\|\hat{\boldsymbol{P}}_{(l)}[:,j]\boldsymbol{M}_{(l)}[j,i]\right\| \leq q_w\delta_w,$$

where $q_w$ is the number of pruned weight entries. Finally, the representation matrix in the $(l+1)$-th layer can be bounded by:

$$\|\hat{\boldsymbol{H}}_{(l+1)} - \boldsymbol{H}_{(l+1)}\|_F \leq \ell_\sigma bc\epsilon\|\boldsymbol{H}_{(l)}\|_F\|\boldsymbol{W}\|_F + \ell_\sigma q_w\delta_w, \tag{17}$$

where $\ell_\sigma$ is the Lipschitz constant representing the non-linearity of activation function $\sigma$ (Virmaux & Scaman, 2018). The above equation corresponds to the one in Proposition 4.4.

The above analysis shows that UNIFEWS with unified graph and weight pruning produces a good approximation of the learned representations across GNN layers, and the margin of output representations is jointly bounded by the graph sparsification rate $\epsilon$ and weight threshold $\delta_w$. A recent work (Zhang et al., 2023b) offers a theoretical evaluation specifically on model weight pruning throughout training iterations, under more narrow assumptions and the particular GCN scheme. We believe their results could be supplemental to our theory whose focus is the graph perspective.

## C. Additional Experiments

### C.1. Detailed Experiment Settings

**Dataset.** To showcase the scalability of UNIFEWS, we evaluate it on a range of datasets where the backbone models are applicable. Statistics of the datasets can be found in Table 3, where we follow the common settings including the graph preparing and training set splitting pipeline in these benchmarks. In the table, we incorporate self-loop edges and count undirected edges twice to better reflect the propagation overhead.

Evaluation are conducted on a server with 32 Intel Xeon CPUs (2.4GHz), an Nvidia A30 (24GB memory) GPU, and 512GB RAM. Specifically, most iterative sparsification baselines fail to produce results on graphs larger than pubmed due to the out of memory (OOM) error. As they usually involve graph-scale trainable matrix or cache, their memory overhead is typically greater than the raw backbone model, rendering them less applicable to large graphs. In comparison, UNIFEWS performs sparsification on the fly and does not incur additional memory footprint.

**Backbone Models.** We select classic GNNs as our backbones, i.e., subject architectures of sparsification methods, due to the consideration that most baselines are only applicable to a couple of classic architectures such as GCN and GAT. In contrast, we demonstrate the generality of UNIFEWS on more backbones including GraphSAGE and GCNII. Iterative backbone models include:

- GCN (Kipf & Welling, 2017) is the representative message-passing GNN with a diffusion matrix $\boldsymbol{T} = \tilde{\boldsymbol{A}}$ across all layers.
- GAT (Brody et al., 2022) learns a variable diffusion $\boldsymbol{T}$ for each layer by multi-head attention. We set the number of heads to 8 for hidden layers.

Table 3: Statistics of graph datasets. $f$ and $N_c$ are the numbers of input attributes and label classes, respectively. Numbers in the "Split" column are percentages of nodes in training/validation/testing set w.r.t. labeled nodes, respectively

| Dataset | Nodes $n$ | Edges $m$ | Features $f$ | Classes $N_c$ | Split |
|---|---|---|---|---|---|
| cora (Kipf & Welling, 2017) | $2,485$ | $12,623$ | $1433$ | $7$ | $0.50/0.25/0.25$ |
| citeseer (Kipf & Welling, 2017) | $3,327$ | $9,228$ | $3703$ | $6$ | $0.50/0.25/0.25$ |
| pubmed (Kipf & Welling, 2017) | $19,717$ | $88,648$ | $500$ | $3$ | $0.50/0.25/0.25$ |
| physics (Shchur et al., 2019) | $34,493$ | $495,924$ | $8415$ | $5$ | $0.50/0.25/0.25$ |
| arxiv (Hu et al., 2020) | $169,343$ | $2,315,598$ | $128$ | $40$ | $0.54/0.18/0.29$ |
| products (Hu et al., 2020) | $2,400,608$ | $123,718,024$ | $100$ | $47$ | $0.08/0.02/0.90$ |
| papers100m (Hu et al., 2020) | $111,059,956$ | $3,228,124,712$ | $128$ | $172$ | $0.78/0.08/0.14$ |

- GraphSAGE (Hamilton et al., 2017) performs more specialized message-passing aggregation and is suitable for large graphs.
- GCNII (Ming et al., 2020) features residual connections and identity mapping. We use $L = 32$ for demonstrating the effect of sparsification on deep GNNs.

For decoupled designs, we implement the pre-propagation version (Wang et al., 2021a) of the following models:

- SGC (Wu et al., 2019) corresponds to GCN in spectral smoothing, but computes the propagation $\boldsymbol{P} = \tilde{\boldsymbol{A}}^L \boldsymbol{X}$ separately in advance.
- APPNP (Klicpera et al., 2019) accumulates and propagates the embedding with a decay factor $\boldsymbol{P} = \sum_{l=0}^{L-1} \alpha(1-\alpha)^l \tilde{\boldsymbol{A}}^l \boldsymbol{X}$.

**Baseline Methods.** Our selection of baselines is dependent on their applicable backbones. We mostly utilize their public source codes and retain original implementations. For iterative models, we consider state of the arts in graph and joint compression, which are methods with the capability to produced sparsified graphs with smaller edge sets for both GNN training and inference:

- GLT (Chen et al., 2021) proposes concurrently sparsifying GNN structure by gradient-based masking on both adjacency and weights.
- GEBT (You et al., 2022) gradually discovers the small model during training. Its implementation is limited to the GCN backbone.
- CGP (Liu et al., 2023a) iteratively prunes and regrows graph connections while exploiting irregular compression on weights.
- DSpar (Liu et al., 2023b) employs one-shot graph sparsification according to a degree-based metric, which implies an upper bound for pruning rate. It does not perform weight pruning.
- Random (RAN) refers to the sparsification method that removes entries with uniform probability based on the specified sparsity.

For the decoupled scheme, we mainly evaluate the graph sparsification. There are two propagation personalization techniques, both only available for the SGC backbone:

- NDLS (Zhang et al., 2021) determines the hop number by calculating the distance to convergence, which produces a customized propagation.
- NIGCN (Huang et al., 2023) offers better scalability by performing degree-based estimation on the node-wise propagation depth.

The graph and weight sparsities are calculated as portions of pruned entries compared to the original matrices. For layer-dependent methods, the average sparsity across all layers is used. For UNIFEWS, the pruning thresholds $\delta_a$ and $\delta_w$ are determined by Theorem 4.2. Note that as Algorithm 1 naturally preserve diagonal elements in $\hat{\boldsymbol{T}}$, a GCN layer with $\eta_a = 100\%$ UNIFEWS pruning is equivalent to an MLP layer.

### C.2. Additional Results on Effectiveness

Extending Figure 3, results of edge and weight sparsification of UNIFEWS and other baselines on more datasets are in Figure 10, while the performance comparison is discussed in Section 5.1. Additionally, we present the experimental results of UNIFEWS on larger datasets and on more backbone architectures in Figure 11 and Figure 12, respectively. No baseline method is available for these settings. It can be observed that UNIFEWS constantly achieves satisfying performance in a large range of sparsity. The effect of graph sparsification on GCNII is relatively unstable, likely because the deeper layers amplify the approximation error.

**Variance.** The variance of UNIFEWS is particularly shown in Figure 13 corresponding to Figure 3. Typically, the variance of iterative UNIFEWS models are between the range of 1-3%, which is slightly larger than the backbone model due to the perturbation of edge and weight sparsification.

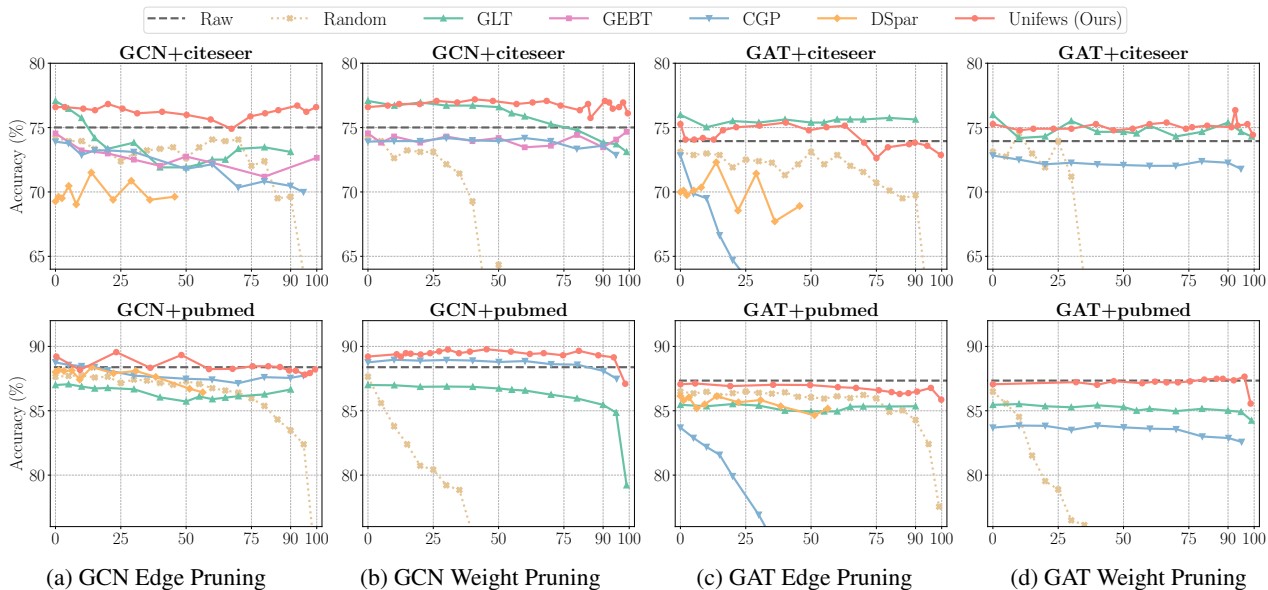

Figure 10: Accuracy of *iterative* models over citeseer and pubmed, supplementing Figure 3. Columns of "Edge" and "Weight Sparsity" present the average results of models with solely edge and weight sparsification, respectively.

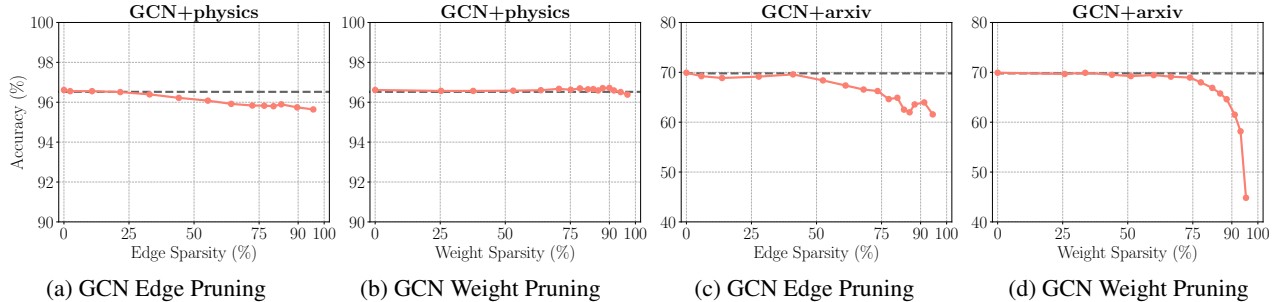

Figure 11: Accuracy of *iterative* UNIFEWS over physics and arxiv, while all baseline methods meet OOM.

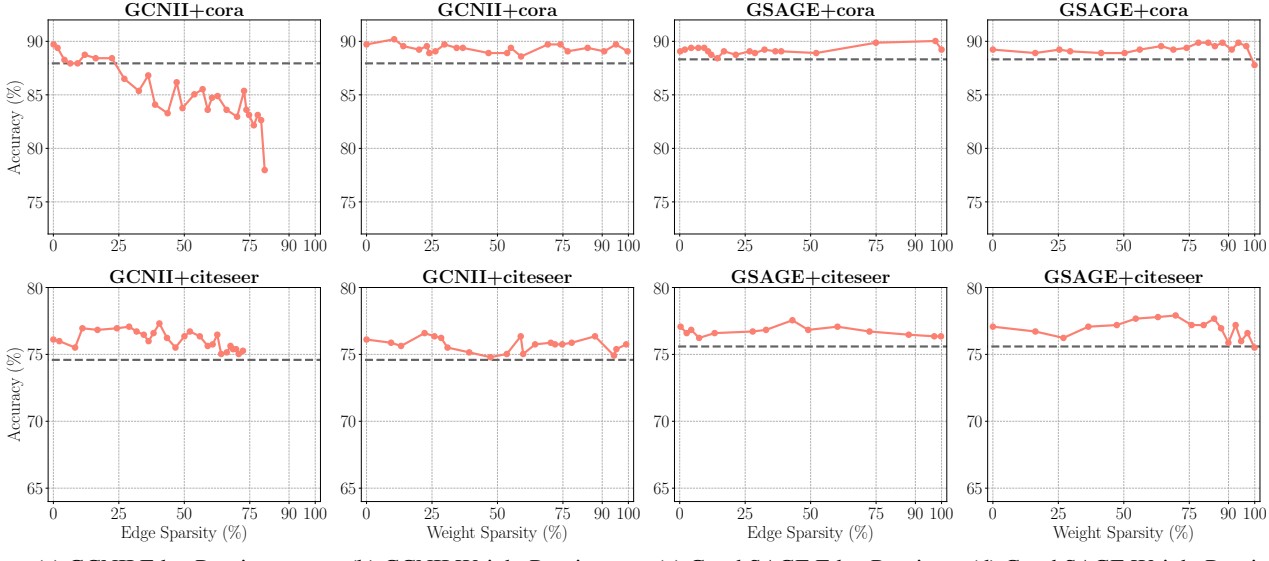

Figure 12: Accuracy of *iterative* UNIFEWS for GCNII and GraphSAGE backbones, while no baseline method is applicable.

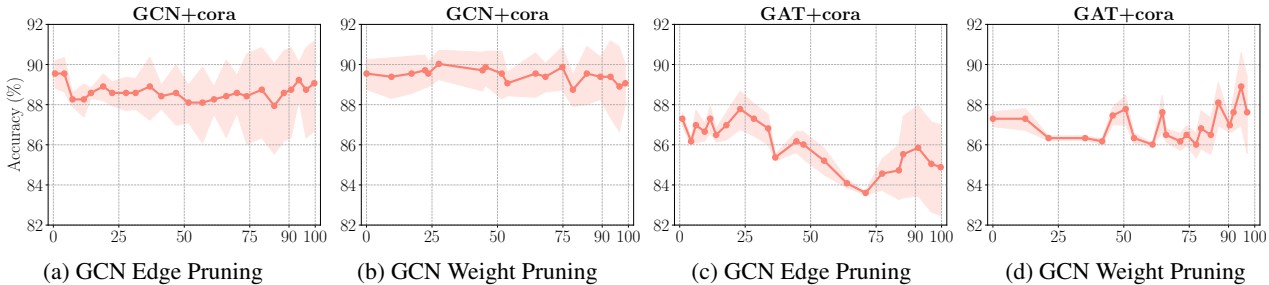

(a) GCN Edge Pruning  (b) GCN Weight Pruning  (c) GCN Edge Pruning  (d) GCN Weight Pruning

Figure 13: Accuracy and variance of *iterative* UNIFEWS over cora.

### C.3. Effect of Hyperparameters

**Effectiveness of Joint Sparsification.** For effectiveness, the impact of jointly changing $\delta_a$ and $\delta_w$ is displayed in Figure 14. Intuitively, GCN is more resilient to UNIFEWS sparsification, considering its relatively high redundancy of the wide distribution of entry values. In comparison, GAT is more sensitive to weight removal, that beyond a certain threshold $\delta_w$, its accuracy drops significantly. This observation suggests that the value of learnable attention weights in GAT are highly concentrated, and selecting an appropriate sparsity is critical to its performance.

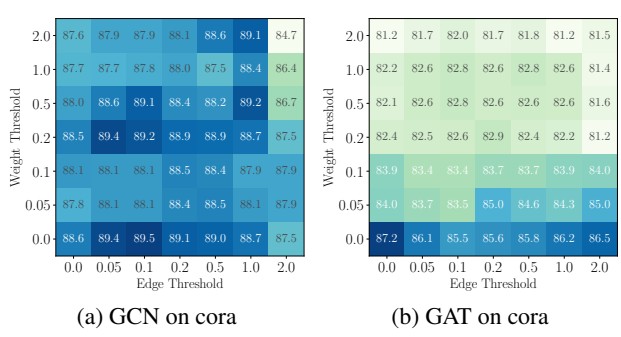

(a) GCN on cora  (b) GAT on cora

Figure 14: Accuracy of varying joint sparsification thresholds on GCN and GAT over cora.

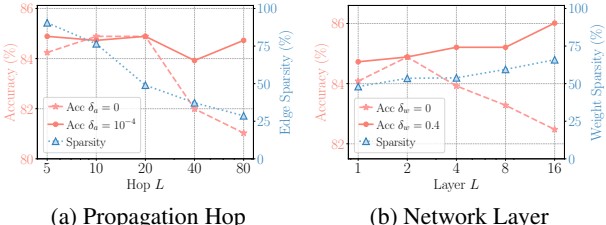

(a) Propagation Hop  (b) Network Layer

Figure 15: Sensitivity of propagation and network transformation layers evaluated on SGC over cora. **(a)** Impact on accuracy and average edge sparsity of varying numbers of propagation hops. **(b)** Impact on accuracy and average weight sparsity of varying numbers of network layers. Note that the x-axis representing layers is on a logarithmic scale.

**Sparsification on Synthetic Graphs.** We further utilize GenCAT (Maekawa et al., 2022; 2023) to generate synthetic graphs with randomized connections and features by varying its parameters. GenCAT$(\alpha, \sigma)$ is the state-of-the-art graph generation approach that allows for configuring the edge connections and node attributes synthesis by specifying their distributions, where $\alpha$ is the coefficient of edge power law distribution $P(d) = d^{-\alpha}$, and $\sigma$ is the deviation of attribute Gaussian distribution $p[v] \sim N(0, \sigma^2)$, both sharing the definitions in our paper. The remaining inputs of GenCAT, including label distribution and attribute-label correlation, are set by mimicking the statistics from cora. We utilize two groups of $\alpha$ and $\sigma$ values to generate 4 graphs with different edge and attribute distributions and evaluate UNIFEWS.

Then, we evaluate the relationship between the edge threshold $\delta_a$ and the edge sparsity $\eta_a$ following the settings of Figure 8. The results are available in Figure 16. As an overview, the pattern is similar to the one in Figure 8, while the constants are effectively affected by the changes of edge and feature distribution in the GenCAT graph. For $\delta_a$ not too close to 0, the relation with $\eta_a$ largely follows Theorem 4.2.

**Propagation and Network Layer $L$.** Since UNIFEWS adopts layer-dependent graphs, we additionally evaluate its performance on models with varying layers. Figure 15(a) is the result of iteratively applying UNIFEWS edge sparsification to SGC with different propagation depths. The backbone model without pruning $\delta_a = 0$ typically suffers from the over-smoothing issue under large hop numbers. On the contrary, as elaborated in Section 4.2, UNIFEWS is powerful for identifying and eliminating unimportant propagations, especially for larger hops, and thereby prevents information loss. With respect to architectural compression Figure 15(b), it is noticeable that UNIFEWS promotes model performance and average weight sparsity at the same time for deeper layers, effectively reducing network redundancy.

### C.4. Specific Experiments

**Entry Distribution.** An example on the real-word graph cora is shown in Figure 9(a), which demonstrates that the distribution of node degree follows the power law, and the distribution of message values is correlated with the diffusion entries. Hence, message magnitude is effective in representing edge importance and determining entry removal. Setting entries in the diffusion matrix $T$ to zero according to Equation (5) implies removing the corresponding edges from the graph diffusion process. Consequently, messages with small magnitudes are prevented from propagating to neighboring nodes and composing the output embedding.

**Approximation Error.** An empirical evaluation of multi-hop UNIFEWS on cora is shown in Figure 9(b), which validates that the approximation error is affected by the sparsification threshold $\delta_a$ as well as propagation hops $L$. Meanwhile, even under aggressive sparsification, the error is still adequately bounded to yield meaningful learning outcomes.

**Over-smoothing.** Additionally, we note that the "error" induced in Algorithm 1 is not necessarily harmful. Examining Equation (1), excessive propagation of common GNNs can lead to a decline in performance known as the *over-smoothing issue*, where the graph regularization is dominant and meaningful node identity is lost (Li et al., 2018). By eliminating a portion of graph diffusion, UNIFEWS effectively alleviates the over-smoothing issue.

We showcase the effect of alleviating over-smoothing in UNIFEWS by depicting the embedding difference to optimization convergence in Figure 9(c). With an increased number of hops, the output embedding tends towards an over-smoothed state. However, a stronger sparsification prevents the rapid smoothing and hereby contributes to better graph learning performance.

We further study the particular effect with the iterative GCNII of 32 layers in Figure 12, and with the decoupled SGC of 5-80 layers in Figure 15. Both results demonstrate that, by increasing model layers, the accuracy of the pruned model is largely preserved thanks to the residual connections. Hence, we conclude that UNIFEWS pruning also benefits accuracy by alleviating over-smoothing.

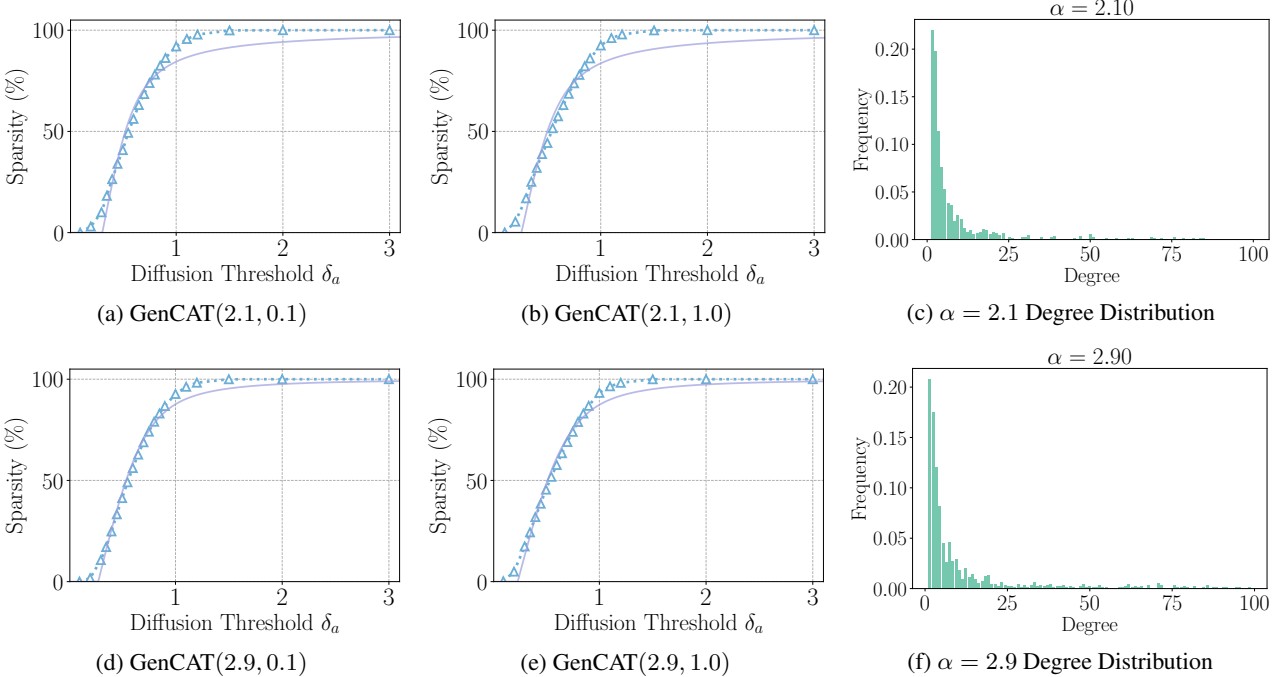

Figure 16: Relationship between edge threshold $\delta_a$ and sparsity $\eta_a$ on synthetic graphs generated by GenCAT. We also present the degree distribution of nodes in different GenCAT graphs controlled by $\alpha$ to illustrate the power law.

