# OpenReview forum: "Unifews: You Need Fewer Operations for Efficient Graph Neural Networks"
_ICML.cc/2025/Conference — ICML 2025 poster_

### Official Review · Reviewer_D2Wc · 2025-03-03

**Overall Recommendation:** 3

**Summary:**

This paper proposes Unifews, a sparsification for both graph and weight matrix. The purpose of such sparsification is to boost the scalability of GNN.

**Claims And Evidence:**

A major claim of this paper is the speed-up on the Ogbn-papers100m dataset, I have some questions about this claim, please refer to latter sections.

**Essential References Not Discussed:**

No.

**Experimental Designs Or Analyses:**

No obvious problem.

**Methods And Evaluation Criteria:**

Yes.

**Other Comments Or Suggestions:**

No obvious typo.

**Other Strengths And Weaknesses:**

S1: The paper is easy to follow.

S2: Theoretical results seem solid.

S3: The literature review is comprehensive.

W1: It would be better to include more baseline. Currently, only graph sparsification methods are used, but since this paper is about scalability, I think other methods should also be included, e.g., other compression method like coarsening/condensation, neighbor sampling methods.

W2: The 100 times acceleration on the ogbn-papers100m is the main contribution of this paper, the authors mention this multiple times in the paper. But the experimental setup of this result is a bit unclear. Please refer to the question section.

**Questions For Authors:**

Q1: About the 100 times acceleration, Table 1 reports 19212s for the graph propagation of SGC on the ogbn-papers100m dataset. It is more than 5 hours. I remember this operation takes roughly 30 minutes on my machine with only CPU, I know there are performance discrepancies among hardwares, but can you double-check your result? It seems too long.

Q2: On the other hand, the acceleration seems conflict with the "Complexity Analysis" section in Line 294? It says the computation complexity is reduced by $O(1-\eta_{\alpha})$, and the $\eta_{\alpha}$ is set to 50% in Table 1, so there should be 2 times of acceleration? This matches the results on all other datasets, but only not ogbn-papers100m. Please kindly correct me if I am missing something.

Q3: DSpar is not used as a baseline in Table 1, could you elaborate why?

Q4: Can you try other scalability method, like GraphSAGE with neighbor sampling, graphsaint and clustergcn on ogbn-papers100m and report the runtime and accuracy? These are all classical scalability methods and their code should be easy to find.

**Relation To Broader Scientific Literature:**

The key contribution is scalability, this is a very active field of GNN.

**Theoretical Claims:**

I briefly check the theoretical claims, they looks solid.

---

> ### Author Rebuttal · Authors · 2025-04-01
>
> We thank the reviewer for the insightful feedback. We have carefully checked the experiments and would like to address the questions as follows.
>
> ## W1&Q4
> We further evaluate other scalability methods and present representative accuracy and training time as below. We also evaluate the smaller `cora` for comparison. Notably, **only `GraphSAGE` is applicable** to `papers100M` through mini-batch training, while other methods incur OOM error. Methods such as `GraphSAINT` and `ClusterGCN` entail longer time for processing the graph data and exhibit weaker scalability on larger graphs.
>
> ### Mini-batch `cora`
> | Edge ratio | 30% | 50% | 80% | 90% |
> |------------|-----|-----|-----|-----|
> | `DropEdge` Acc (%) | 86.4  | 86.8 | 79.5 | 43.7 |
> | `DropEdge` Time (s) | 7.13 | 7.31 | 7.09 | 6.87 |
> | `GraphSAGE` Acc (%) | 77.6 | 78.2 | 79.4 | 78.7 |
> | `GraphSAGE` Time (s) | 2.00 | 1.80 | 1.86 | 1.83 |
> | `GraphSAINT` Acc (%) | 74.7 | 78.1 | 77.8 | 77.2 |
> | `GraphSAINT` Time (s) | 228.02 | 167.29 |  179.58 | 193.31 |
> | `ClusterGCN` Acc (%) | 78.3 | 76.9 | 77.1 | 74.3 |
> | `ClusterGCN` Time (s) | 4.50 | 4.20 | 3.47 | 2.66 |
>
> ### Mini-batch `papers100M`
> | Edge ratio | 30% | 50% | 80% | 90% |
> |------------|-----|-----|-----|-----|
> | `GraphSAGE` Acc (%) |  46.25   |  48.45   |  41.02   |  42.64   |
> | `GraphSAGE` Time (s) |  18208   |  13662   |  6565   |  3850   |
>
> We would like to highlight the difference between the settings and our evaluation in the paper as follows:
>
> 1. The training is conducted in **mini-batch** with batch size=100K. Testing is performed in full-graph manner on CPU due to the excessive graph size.
>
> 2. The "edge ratio" is calculated by $\eta=1-m'/m$, where $m'$ is the total number of sampled neighbors used in propagation, and $m$ is the edge size in the raw graph. This concept is different from edge sparsity since the **propagation graph is not static**.
>
> 3. As elaborated in *Sec A.2*, the graph structure used for coarsening and sampling methods is dynamically determined during training, which limits their wider generality. For example, they cannot be described by the graph smoothing framework, and **can hardly apply to decoupled models**.
>
> ## W2&Q1
> We elaborate the setups of the `papers100m` result in *Tab 1*:
>
> 1. We find that the previous result of `SGC` is largely affected by the "**cold start**" issue, where the first run is significantly slower than the others due to the overhead in accessing and loading the data. By excluding the first run, the average time is around 15K seconds, which is close to `APPNP` and `GraphSAGE`.
>
> 2. To ensure comparable results with baselines, the decoupled propagation of all methods in Tab 1 is implemented by **single-thread C++ computation** that successively performs message-passing for each edge. The process may be slower than the common implementation based on matrix computation, such as Eigen in C++ and PyTorch/Numpy in Python.
>
> 3. The wall-clock time also includes auxiliary operations within the propagation, such as **accessing the storage of attributes and neighboring nodes**. Our implementation stores edges in the format of an adjacency list, which is tailored for the pruning operation. However, it may be slower than the matrix format when the edge scale is large, and results in a greater scalability issue for the full `papers100m`.
>
> In practice, both the baseline and Unifews computation can be accelerated by approaches such as multi-threading, vectorization, and enhanced data structures. However, the relative speedup can still be achieved considering the FLOPs reduction.
>
> ## Q2
> The acceleration ratio of wall-clock time is also affected by several factors beside the complexity analysis:
>
> 1. As suggested in *Sec 4.3*, the complexity provides a lower bound for computation reduction. As the **edges removal is accumulated** across hops, the actual reduction rate can be larger than the theoretical bound. In fact, Unifews achieves $3.3\times$ FLOPs reduction compared to SGC, which is greater than the expected $2\times$.
>
> 2. Evaluation in *Fig 7* shows the **improvement of embedding sparsity** brought by Unifews pruning. This effect is more significant for sparse graphs such as `papers100m`, since the magnitude of node embedding is more likely to diminish when edges are pruned. The improved sparsity further facilitates pruning in subsequent layers and leads to faster computation.
>
> 3. As elaborated in *Q1(3)*, the overhead of **auxiliary graph operations** is more sensitive to the graph scale. For example, finding a neighbor of a node is faster if the edge set is smaller. This may result in speedup greater than the linear improvement.
>
> ## Q3
> In *Tab 1*, we only present the results of approaches applicable to **decoupled models** with the separated graph propagation in C++. Since the official implementation of `DSpar` is based on the PyG framework for iterative models, we are unable to apply it to the decoupled propagation.

---

### Official Review · Reviewer_9jfo · 2025-03-12

**Overall Recommendation:** 4

**Summary:**

This paper introduces UNIFEWS a technique to sparsify GNNs by dropping messages. The authors justify their UNIFEW by proofs that provide theoretical guarantees. In experiments, UNIFEWS allows dropping almost all edges in the graphs without much impact on predictive performance. This leads to a significant speed-up by a factor of up to x100.

## update after rebuttal
The authors have addressed many concerns effectively in the rebuttal. While I still believe the work primarily demonstrates that graph structure is often unimportant for many datasets, the authors' contributions are nonetheless meaningful and well-supported. My recommendation lies between a weak accept and an accept; given the options, I lean toward recommending acceptance.

**Claims And Evidence:**

The authors provide convince evidence in the form of proofs and experimentts.

**Essential References Not Discussed:**

No.

**Experimental Designs Or Analyses:**

The experimental design seems sound.

**Methods And Evaluation Criteria:**

See (W2).

**Other Comments Or Suggestions:**

- Please use `\mid` in your definition of neighborhood instead of |  ($\{ a | ...\}$ vs  $\{ a \mid ...\}$)
- "in an one-time", maybe "a one-time"?
- Theorem 4.1 could be formulated better, it is not mentioned how the sparsified laplacian is obtained which is the crucial part of the theorem.

**Other Strengths And Weaknesses:**

**Strengths:**
- (S1)The proposed method is intuitive and conceptually simple: it is clear that dropping unimportant messages will lead to a speed-up at little cost to predictive performance (the hard part is chosing which messages to drop).
- (S2) The proposed method is backed by theoretical arguments:
	- Theorem 4.1 gives us a bound on the spectral similarity between sparsified and non-sparsified laplacian
	- For decoupled architectures, UNIFEWS allows for an $\epsilon$ approximation of graph smoothing (Proposition 4.2)
- (S3) The proposed method allows to drop almost all edges without losing predictive performance (Figure 4).

**Weaknesses:**
- Speed Analysis (Section 5.2) is done only for decoupled models and not for iterative models. Since decoupled models are more niche than iterative models, this weakens the paper. This is especial crucial as Fig. 6 shows that for iterative models the propagation step is not important, implying that UNIFEWS might lead to only little speedup on iterative models.
- (W2) I am not convinced that the strong predictive performance in the low edge regime (S2) is truly caused by UNIFEWS. After all, achieving the same accuracy with 1% with just edges should simply mean that the graph structure is uninformative. In this case, it is not clear what the true utility of UNIFEWS is. This could be combated, by testing UNIFEWS on more diverse datasets to see how it behaves for problems where the graph structure is important

**Overall,** I think that this is a good paper that is held back by some small issues (mainly W2). Thus I vote for weak accept.

**Questions For Authors:**

- The operation performed by UNIFEWS simply thresholds out messages (=vectors) whose magnitude is smaller than some $\delta$. This operation is by default non-differentiable. Could this cause problems? Might it improve to make this process differentiable during training?

**Relation To Broader Scientific Literature:**

Related work is good.

**Theoretical Claims:**

I did not check the proofs.

---

> ### Author Rebuttal · Authors · 2025-04-01
>
> We are thankful for the detailed and insightful comments from the reviewer. We address the specific reviews below with further evaluation results.
>
> ## W1
> In the paper, we do not show the wall-clock time for iterative models mainly because of the **variety in baseline implementations**. The existing approaches vary greatly in terms of GNN frameworks and the implementation of sparsification. For example, `GLT` applies masks on the adjacency matrix, while `CGP` utilizes learnable edge weights stored in variable tensors. As a result, the wall-clock time among different models is hardly comparable in a fair manner. We hence employ FLOPs to measure the expected computation cost, which is also widely used in related studies.
>
> Furthermore, the FLOPs measured in *Fig 6* of different operations are not equivalent to wall-clock times. This is because the **feature transformation is performed on GPU** with highly efficient dense-dense matrix-matrix multiplication (MM), while the graph propagation as sparse-dense matrix-matrix multiplication (SPMM) is not applicable to most acceleration techniques. In fact, the adjacency matrix is usually loaded on CPU, rendering the propagation a complicated cross-device and irregular computation, which is usually considered as the bottleneck of GNN efficiency.
>
> We present the inference time breakdown of Unifews for `GCN+arxiv` in the following table. It can be observed that both operations benefit from the sparsification.
>
> ### `GCN+arxiv`
> | Edge Sparsity | Weight Sparsity | Graph Propagation (s) | Feature Transform (s) |
> | --- | --- | --- | --- |
> | 0% | 0% | 5.85 | 0.58 |
> | 50% | 0% | 3.54 | 0.61 |
> | 0% | 50% | 5.54 | 0.36 |
> | 50% | 50% | 3.66 | 0.36 |
>
> In summary, we believe achieving practical speedup is largely an *implementation* problem. There is a series of works [Deng et al., 2020a] developing software and hardware systems and achieving real speed improvement, which is orthogonal to our contribution on how to achieve such sparsity by *algorithmic* design.
>
> ## W2
> While approaching 100% sparsity, Unifews still **preserves certain graph structural information**, which is critical to model performance. For comparison, we present the result of `MLP` with no graph information as below. It can be observed that the performance improvement is significant on `cora` and `citeseer`.
>
> ### `MLP` ($L=2$)
> | Dataset | `cora` | `citeseer` | `pubmed` |
> | --- | --- | --- | --- |
> | Acc (%) | 75.2 | 71.7 | 87.0 |
>
> We would like to discuss the difference of Unifews with MLP at high sparsity in the following aspects:
>
> 1. Given the large number of edge/weight entries, a number of entries remain even when the pruning threshold is high. As a result, the pruning ratio rarely reaches exactly 100% in our experiments. This phenomenon has also been observed in previous NN and GNN pruning studies [Liu et al., 2023a], where maintaining **a small fraction (0.1%-1%) of edges or weights can largely preserve GNN performance**.
>
> 2. As discussed in *Fig 7*, Unifews brings higher sparsity to the learned representation, which is known to be **beneficial for model learning**. This improvement is observed for both edge and weight sparsification in previous works such as [You et al., 2022; Chen et al., 2021]. As the sparsity is progressively acquired during training iterations, the model benefits from learning on perturbed variants of the data, leading to improved performance.
>
> 3. Due to the implementation of acquiring the **normalized adjacency $\tilde{A}$** in PyG, the diagonal entries are $D^{-1}$ instead of $I$. Consequently, when the edge sparsity is close to 100%, each graph propagation can be viewed as a normalization to node features based on degrees, which is different from MLP. This process also incorporates graph structural information and may also contribute to better performance.
>
> ## C1-3
> We thank the reviewer for pointing out the issues. We will carefully fix the typos in the revised version.
>
> The sparsified $\hat{L}$ corresponds to the pruned edge set $\hat{\mathcal{E}}$ acquired by Unifews as in *Lemma 3.3* and *Lemma B.1*. We will improve the formulation in the revised version.
>
> ## Q1
> The pruning process itself is **not differentiable**, similar to conventional neural network pruning [Han et al, 2015]. Intuitively, pruning can be implemented by applying a 0-1 mask to the target matrix every time it is used during forward inference and backward propagation. Gradients of the pruned entries (i.e., with zero values) are naturally kept as zero. Hence, the pruning process **does not affect normal model training**.
>
> It is possible to augment the pruning to be differentiable, or even adaptive during training. Works such as AdaptiveGCN, SGCN, and Shadow-GNN discussed in *Sec A.1* are similar to this idea. However, the process may **incur additional overhead** for learning these variables. Hence, in this paper, we mainly use the simple static strategy to ensure efficiency and align with our theoretical analysis.

---

> > ### Comment · Reviewer_9jfo · 2025-04-04
> >
> > Thanks for the reply. I think you raise some good points. However, I remain unconvinced whether your results are not simply showing that GNNs are the wrong tool for the given datasets. Especially, W2.3 is an interesting direction that might be worth investigating in the future. I chose to maintain my score.

---

> > > ### Author Response · Authors · 2025-04-05
> > >
> > > We thank the reviewer for the feedback. To further investigate the performance with high sparsity and with diagonal entries, we present ablation studies comparing MLP, unpruned GCN, and Unifews with different diagonal schemes. We also extend to the heterophilic datasets `cornell` and `wisconsin` [R1], where the graph information is known to be malignant to non-heterophilic models such as GCN. In the following table, names in the format of `Unifews-99-0` denote Unifews with edge sparsity 99% and weight sparsity 0%, while `Unifews(0)` and `Unifews(1)` refer to the variants by setting the diagonal entries of the adjacency matrix to 0 and 1, respectively. Results better than the unpruned GCN are highlighted in **bold**. We mainly draw the following conclusions:
> > > 1. The training process of `Unifews` is similar to `GCN`, showing similar patterns of performance improvement or degradation compared to `MLP`. This indicates that even when a large portion of the edges is pruned, the graph structure can still be gradually learned during the training iterations through message passing based on the remaining edges.
> > > 2. The diagonal entries, i.e., using $A+I$ instead of $A$ to represent the graph structure, are critical for model performance, which is consistent with the GCN paper [Kipf & Welling, 2017]. Hence, `Unifews(0)` usually performs worse and is particularly poor with high edge sparsity.
> > > 3. The effect of the diagonal entries can be viewed as **amplifying the inductive bias from graph information** when training with the pruned graph. If the graph structure is benign (`cora` and `citeseer`), pruning while keeping the diagonal entries may further improve the performance. Conversely, keeping the diagonal entries further decreases the accuracy under heterophily (`cornell` and `wisconsin`).
> > > 4. Whether `Unifews` (diagonal $D^{-1}$) or `Unifews(1)` (diagonal $I$) is better depends on the specific dataset. We hence use the former one to be consistent with the PyG GCN implementation.
> > >
> > > | Dataset | `cora` | `citeseer` | `pubmed` | `cornell` | `wisconsin` |
> > > |-|-|-|-|-|-|
> > > | `MLP` | 75.2 | 71.7 | 87.0 | 73.0 | 80.4 |
> > > | `GCN` | 88.3 | 74.9 | 88.8 | 59.5 | 64.7 |
> > > | `Unifews-50-0` | **89.3** | **76.0** | 87.9 | 59.5 | 56.9 |
> > > | `Unifews(0)-50-0` | 86.5 | 71.6 | 83.8 | 59.4 | 51.0 |
> > > | `Unifews(1)-50-0 `| **88.4** | **75.1** | 88.0 | 59.5 | 56.9 |
> > > | `Unifews-99-0` | **89.1** | **76.0** | 88.1 | 43.2 | 52.9 |
> > > | `Unifews(0)-99-0` | 45.5 | 30.5 | 49.8 | 43.2 | 52.9 |
> > > | `Unifews(1)-99-0` | **88.9** | **75.5** | 88.5 | 40.5 | 52.9 |
> > >
> > > [R1] Geom-GCN: Geometric Graph Convolutional Networks. ICLR'20.

---

### Official Review · Reviewer_t1S7 · 2025-03-12

**Overall Recommendation:** 3

**Summary:**

This paper explores strategies to accelerate GNN computation by integrating both structural sparsification and weight parameter pruning. Specifically, it introduces a framework called UNIFEWS, which adaptively and progressively simplifies computations while providing theoretical guarantees on the accuracy tradeoff. Experimental results validate the proposed approach.

**Claims And Evidence:**

Yes.

**Essential References Not Discussed:**

N/A

**Experimental Designs Or Analyses:**

I find the experimental section to be quite comprehensive and did not notice any major shortcomings, except for a few questions, which I have outlined in the "Questions" section below.

**Methods And Evaluation Criteria:**

Yes.

**Other Comments Or Suggestions:**

Please see my questions below.

**Other Strengths And Weaknesses:**

## Strengths

1. The paper is well-presented with excellent readability, supported by clear diagrams and algorithmic aids for better clarification.
2. The distinction between iterative GNNs and decoupled GNNs enhances the generality of the proposed approach.
3. A complexity analysis of the proposed method(s) is provided.

## Weaknesses

1. DropEdge [Rong et al., 2020] should be included as a baseline in the comparisons for iterative sparsification.
The variance of numerical metrics should be reported in all plots and tables.
2. Please check my questions below.

**Questions For Authors:**

## Questions
1. In the experiments, the backbone GNNs are all set to 2 layers, effectively avoiding the over-smoothing problem. This suggests that the improvements from sparsification may stem from factors other than mitigating over-smoothing. What are your thoughts on this? What other possible explanations could there be?
2. The results in Figures 3 and 5 are somewhat difficult to interpret when weight/edge sparsity approaches 100%, as the performance even improves compared to the corresponding backbones. Wouldn’t a 100% edge sparsity reduce the model to an MLP, making performance entirely dependent on node features? Similarly, when weight sparsity nears 100%, no node feature information should be preserved—so how is high node classification accuracy still achieved?

**Relation To Broader Scientific Literature:**

The broader connection of this paper to general scientific discovery is not immediately clear. However, the proposed efficiency improvements contribute to the wider deployment of GNNs on larger-scale datasets and applications, which is certainly a valuable advantage.

**Theoretical Claims:**

For Theorem 4.1, I have not verified the proof; however, it is crucial to include synthetic examples illustrating the bounds. This would help readers better understand their behavior, especially since the ranges of the constants in the theorem are not provided.

---

> ### Author Rebuttal · Authors · 2025-04-01
>
> We sincerely appreciate the constructive feedback from the reviewer. Below, we provide detailed responses with new experiments following the suggestions.
>
> ## T1
> The range of the constants in *Thm 4.1* is discussed in *Sec B.2*. The constant $2<\alpha<3$ represents degree distribution, and $\sigma>0$ is the standard deviation of feature distribution. As $t=1/(\alpha-1)$, there is $0.5<t<1$. The constant $C>0$ is related to the norm of the embedding $\|p\|$, as it maps the sparsity ratio $0\le\eta\le 1$ to the actual threshold value applied to the embedding.
>
> On **realistic datasets**, we perform evaluation in *Fig 12*, where the blue line shows that by varying the threshold $\delta_a$ (x-axis), the acquired sparsity $\eta_a$ (right y-axis) aligns well with the relationship in *Thm 4.1* with two dataset-specific constants.
>
> We further utilize **GenCAT** [R1,R2] to generate synthetic graphs with randomized connections and features by varying its parameters. Then, we evaluate the relationship between the edge threshold $\delta_a$ and the edge sparsity $\eta_a$ following the settings of Fig 12. The results are available in: https://anonymous.4open.science/r/Unifews-A91B/plot.pdf . As an overview, the pattern is similar to the one in Fig 12, while the constants are effectively affected by the changes of edge and feature distribution in the GenCAT graph.
>
> [R1] GenCAT: Generating attributed graphs with controlled relationships between classes, attributes, and topology. Information Systems'23.
> [R2] Beyond Real-world Benchmark Datasets: An Empirical Study of Node Classification with GNNs. NeurIPS'22.
>
> ## W1
> We present the results of `DropEdge` for `GCN+cora` as follows. As mentioned in the DropEdge paper, its scheme is **different from graph sparsification** since: (1) its edge removal is performed randomly at each training time; (2) the dropped edges are not accumulated throughout propagation layers, which differs from Unifews. In comparison, the `Random` baseline used in our experiments is closer to Unifews regarding the two aspects above, while using random edge removal. Hence, we mainly use the `Random` baseline in our experiments.
>
> |Edge Sparsity|30%|50%|80%|90%|
> |--|--|--|--|--|
> |`DropEdge`|86.4|86.8|79.5|43.7|
> |`Random`|85.5|86.3|85.0|80.8|
>
> Empirically, the **variance** of Unifews is within the range of 1-3%, which is slightly larger than the backbone model due to the perturbation of edge and weight sparsification. We will include the variance in the revised paper.
>
> ## Q1
> For iterative models, Unifews removes insignificant entries and improves representation sparsity as shown in *Fig 7*. The **increased sparsity** is known to be beneficial for model learning, as revealed by [Han et al., 2015] for general neural networks, and similarly evaluated for GNNs in [You et al., 2022; Chen et al., 2021]. In brief, it is believed that sparsification can be viewed as a form of perturbation during learning. By enhancing sparsity, the model can strengthen the useful neural connections within network weights and focus on the most informative features, leading to improved performance.
>
> Regarding **over-smoothing**, we further study the particular effect with the iterative `GCNII` of 32 layers in Appendix *Fig 11*, and with the decoupled `SGC` of 5-80 layers in *Fig 14*. Both results demonstrate that, by increasing model layers, the accuracy of the pruned model is largely preserved thanks to the residual connections. Hence, we conclude that Unifews pruning also benefits accuracy by alleviating over-smoothing.
>
> ## Q2
> The performance near 100% sparsity is affected by various factors. We would like to compare it to MLP in the following aspects:
>
> 1. Given the large number of edge/weight entries, a number of entries remain even when the pruning threshold is high. As a result, the pruning ratio rarely reaches exactly 100% in our experiments. This phenomenon has also been observed in previous NN and GNN pruning studies [Liu et al., 2023a], where maintaining **a small fraction (0.1%-1%) of edges or weights can largely preserve performance**.
>
> 2. As elaborated in *Q1*, **higher sparsity** induced by Unifews pruning can inherently enhance model performance. During training iterations where pruning is progressively applied, the model benefits from learning on perturbed variants of the data, leading to improved performance. This improvement is observed in both edge and weight sparsification.
>
> 3. Due to the implementation of acquiring the **normalized adjacency $\tilde{A}$** in PyG (`torch_geometric.nn.conv.gcn_conv.gcn_norm`), the diagonal entries are $D^{-1}$ instead of $I$. Consequently, when the edge sparsity is close to 100%, each graph propagation can be viewed as a normalization to node features based on degrees, which is different from MLP. This process incorporates graph structural information and may also contribute to better performance. We will revise *Alg 1* to reflect this distinction explicitly.

---

> > ### Comment · Reviewer_t1S7 · 2025-04-03
> >
> > Dear authors,
> >
> > Thank you for your rebuttal. I find the additional experiments and explanations quite helpful, and the comparisons with DropEdge provide valuable insights. I will keep my score, since it's already positive. However, regarding Q2.3, if $A$ is already $\mathbf{0}$, how do the aggregation and propagation operators retain any structural information? I might missed something, or please verify and clarify this part accordingly.
> > Best,

---

> > > ### Author Response · Authors · 2025-04-05
> > >
> > > We thank the reviewer for the comments. We would like to elaborate on the scheme of diagonal entries in the adjacency matrix with more details. As in Sec 2.1, we use the **self-looped adjacency matrix $\bar{A} = A + I$** to represent the graph structure. The diagonal entries can be regarded as residual connections for keeping the node features during propagation. During Unifews pruning, the diagonal entries are naturally preserved, which is represented by line 3 in Alg 1: $\hat{P}\_{(l+1)} \gets \hat{P}_{(l)}$. These entries are excluded in sparsity calculation, since they do not cost additional computation.
> > >
> > > In Q2.3, we intend to elaborate that, due to the graph normalization implementation, the actual propagation matrix used in `Unifews` for iterative GCN is $\tilde{A} = D^{-1/2}(A + I)D^{-1/2}$. In this case, Alg 1 line 3 should be modified to $\hat{P}\_{(l+1)} \gets D^{-1}\hat{P}_{(l)}$, which still preserves certain structural information when the rest of $A$ is pruned to 0.
> > >
> > > Another possible alternative is setting diagonal entries outside normalization: $\tilde{A} = D^{-1/2}A D^{-1/2} + I$ (denoted as `Unifews(1)`), which is equivalent to MLP when $A=O$. We conduct additional experiments to compare the two schemes in the reply to *Reviewer 9jfo*. In summary, the results imply that the diagonal entries, along with other edges, can **amplify** (not necessarily **improve**) the inductive bias from graph information during Unifews training. Hence, the improvement over the unpruned backbone is possibly a special case on certain datasets.

---

### Official Review · Reviewer_mKp3 · 2025-03-14

**Overall Recommendation:** 3

**Summary:**

The paper proposes a framework named UNIFEWS (UNIFied Entry-Wise Sparsification), which aims to improve the learning efficiency of Graph Neural Networks (GNNs) by jointly sparsifying the graph and the weight matrix. By incrementally increasing sparsity layer by layer, the framework significantly reduces the computational operations in GNNs without notably compromising model accuracy. Theoretically, the authors establish a new framework to characterize the learning of sparsified GNNs and prove that UNIFEWS can effectively approximate the learning objective within bounded error. Experiments show that UNIFEWS achieves efficiency improvements on multiple datasets, reducing matrix operations by 10 to 20 times and accelerating computations by up to 100 times for graphs with billions of edges.

**Claims And Evidence:**

Yes.

**Essential References Not Discussed:**

No.

**Experimental Designs Or Analyses:**

Yes. All experimental frameworks were rigorously validated through specific statistical method, and cross-verified against benchmark datasets, documented in Evaluation Section and Supplementary Materials.

**Methods And Evaluation Criteria:**

Yes.

**Other Comments Or Suggestions:**

The paper does not discuss the limitations of the proposed method.

**Other Strengths And Weaknesses:**

Strength:
* A novel joint sparsification technique is proposed, which unifies the operations of the graph and the weight matrix and establishes a theoretical connection between the graph optimization process and sparsification. This innovative perspective provides new ideas for the optimization of GNNs and fills the theoretical gap in existing research regarding the joint sparsification of graphs and weights.
* The concept of ϵ-spectral similarity is introduced, and its effectiveness in the spectral domain is theoretically demonstrated for UNIFEWS. This provides a more rigorous theoretical guarantee for the sparsification of GNNs.
* It demonstrates significant efficiency improvements on large-scale graph data, especially achieving a 100-fold acceleration on graphs with billions of edges. This is of great significance for processing large-scale graph data and can effectively alleviate the bottleneck issues of existing GNNs in terms of computational resources and time.
* By incrementally increasing sparsity layer by layer, UNIFEWS progressively reduces the computational load in multi-layer GNNs, further enhancing the scalability of the model.

Weakness:
* Although the paper proposes a theoretical framework to analyze the impact of sparsification on GNN learning, its theoretical analysis is based on several assumptions, such as the distribution of the graph (e.g., power-law distribution) and the Gaussian distribution of input features. These assumptions may not always hold in practical applications, which could lead to certain discrepancies between the theoretical results and the actual performance. Please further clarify the rationality.
* The performance of UNIFEWS depends on the choice of sparsification thresholds (δa and δw). However, the authors do not provide a systematic method for automatically selecting these hyperparameters, instead relying on manual tuning or empirically based choices. This may make it difficult to find the optimal combination of hyperparameters in practical applications, thereby affecting the performance of the model.

**Questions For Authors:**

Please refer to the Other Strengths And Weaknesses section for detailed inquiries.

**Relation To Broader Scientific Literature:**

This study makes a meaningful contribution to the existing body of knowledge in the related literature.

**Theoretical Claims:**

Yes.

---

> ### Author Rebuttal · Authors · 2025-04-01
>
> We are thankful to the reviewer for recognizing our theoretical and experimental contributions. We respectfully address the specific reviews below.
>
> ## W1
> **Power-law** for degree distributions is frequently observed in real-world graphs, especially large-scale ones focused on in this study, such as social networks, citation networks, and web graphs [R1]. We present an empirical evaluation in *Fig 8(a)*, where the blue bars show the distribution of inversion of edge degrees on Cora. It can be observed that edges with larger magnitudes (i.e., smaller degrees) exhibit higher relative density, while only a small portion of edges have magnitudes close to 0 (i.e., nodes with large degrees).
>
> The assumption of the **Gaussian distribution** can apply to node attributes or output features of linear transformations, depending on the exact input of iterative or decoupled models. The Gaussian distribution is commonly used for depicting the feature distribution of neural networks as an extension of the central limit theorem. For example, text embeddings are commonly used as node attributes on text-attributed graphs, which can be regarded as multivariate Gaussian distributions. The similar assumption is commonly employed in graph generation and GNN research [R2-R5].
>
> We will clarify the rationale when deriving Thm 4.1 in the revised version.
> In *Reviewer t1S7 T1*, we further present a new empirical evaluation regarding these two assumptions on synthetic graphs generated by GenCAT.
>
> [R1] Power-law distributions in empirical data. SIAM Rev'09.
> [R2] Contextual Stochastic Block Models. NeurIPS'18.
> [R3] Deep Gaussian Embedding of Graphs: Unsupervised Inductive Learning via Ranking. ICLR'18.
> [R4] Distribution Knowledge Embedding for Graph Pooling. TKDE'22.
> [R5] GenCAT: Generating attributed graphs with controlled relationships between classes, attributes, and topology. Information Systems'23.
>
> ## W2
> We mainly utilize *Thm 4.1* for determining the **edge threshold $\delta_a$**. In realistic applications for GNN sparsification, the sparsity $\eta_a$ is usually given by users. Then, as indicated by Thm 4.1, $\delta_a$ is monotonically related to $\eta_a$ with two constants determined by the dataset and model aggregation. To set the threshold for a model-dataset pair, only a few trial epochs under different $\delta_a$ are needed to fit the actual curve between $\delta_a$ and $\eta_a$ and decide the desired value of $\delta_a$.
> Empirical evaluation in *Fig 12* (blue line) shows that the relationship between $\eta_a$ (right y-axis) and $\delta_a$ (x-axis) aligns well with the theory. In fact, for a large range of sparsity (empirically 10%-90%), the relationship is almost linear (note that Fig 12 is drawn with a logarithmic x-axis), which further simplifies the selection.
>
> The **weight threshold $\delta_w$** can be similarly determined by fitting empirical trials when additionally assuming a Gaussian distribution for the weight matrix. This process is identical to conventional neural network pruning in practice, such as [Han et al, 2015].
>
> ## C1
> One potential limitation and future direction lies in the consideration of graph heterophily, as the current strategy prunes insignificant messages. However, under heterophily, where connected nodes have dissimilar labels, messages with large magnitudes may not be beneficial to model inference. In this case, the graph smoothing objective *Def 3.1* needs to be refined, and consequently, the sparsification strategy should be adjusted. While there are recent works on the spectral optimization process for heterophilic graphs, how to apply sparsification is largely unexplored.

---

### Decision · Program_Chairs · 2025-05-01

**Decision:**

Accept (poster)

**Comment:**

The reviewers unanimously recommend acceptance of the paper with varying degrees of strength. I agree with their assessment and am happy to recommend acceptance of this paper. Several reviewers mentioned that the paper is clear and well-written. They furthermore appreciated the theoretical backing of the proposed method as well as the close to 100-fold acceleration that is observed in one instance. The reviewer's concerns have largely been addressed in the discussion period.

The reviews and the rebuttal have given rise to several interesting points and results that I encourage the authors to include in their revised manuscript. This includes the missing discussion of the limitations of our method, especially with regards to heterophilic graphs, as well as the additional empirical results you provided during the rebuttal.